# A mixed studies systematic review on the health and wellbeing effects, and underlying mechanisms, of online support groups for chronic conditions

Freya Mills [1,2] ✉, John Drury [2], Charlotte E. Hall [1,3], Dale Weston [1], Charles Symons [1], Richard Amlôt [1] & Holly Carter [1]

This pre-registered systematic review aimed to examine whether online support groups affect the health and wellbeing of individuals with a chronic condition, and what mechanisms may influence such effects. In September 2024, literature searches were conducted across electronic databases (Medline, Embase, PsycInfo, Web of Science and Google Scholar), pre-publication websites (MedRxiv and PsyArXiv) and grey literature websites. Qualitative and quantitative studies were included if they explored the impact of online support groups on the health and wellbeing outcomes of individuals with a chronic condition. The Mixed Methods Appraisal Tool was used to appraise the quality of the included studies. In total 100 papers met the inclusion criteria with their findings presented in a thematic synthesis. Health and wellbeing outcomes were categorised as: physical health, mental health, quality of life, social wellbeing, behaviour and decision-making, and adjustment. Mechanisms reported in these studies related to exchanging support, sharing experiences, content expression, and social comparison. User and group characteristics were also explored. The included studies suggest that online support groups can have a positive impact on social wellbeing, behaviour, and adjustment, with inconclusive findings for physical health and quality of life. However, there is also the possibility of a negative effect on anxiety and distress, particularly when exposed to other group members' difficult experiences. Research comparing different online group features, such as platforms, size, and duration is needed. In particular, future research should be experimental to overcome the limitations of some of the cross-sectional designs of the included studies. The review was funded by the National Institute for Health and Care Research Health Protection Research in Emergency Preparedness and Response. Pre-registration ID: CRD42023399258

Chronic conditions refer to health problems that require ongoing management over a period of years that cannot currently be cured, but can be controlled[1]. Almost half of the UK population reported living with at least one long-standing health problem in 2020[2], and globally 41 million people per year are estimated to die from a chronic condition[3]. Although more recent data on the prevalence of chronic conditions is unavailable, it is likely to have increased since the COVID-19 pandemic, with nearly 2 million people reporting symptoms of Long Covid in England and Scotland in March 2023[4]. Living with a chronic condition is associated with reduced health-related quality of life[5,6] and leaves many individuals unable to carry out day-to-day activities, socialise or work, which can result in them being dependent on other people[7,8].

Alongside experiencing symptoms of a chronic condition, individuals may face challenges such as prejudice[9], stigma[10] and feeling alone[8]. One way in which individuals can connect with others, and find support, is through online support groups. Online support groups, also referred to as 'online

[1]Behavioural Science and Insights Unit, UK Health Security Agency, Salisbury, UK. [2]School of Psychology, University of Sussex, Falmer, UK. [3]Department of Psychological Medicine, Kings College London, London, UK. ✉e-mail: freya.mills@ukhsa.gov.uk

communities', 'online support forums', and 'virtual support groups', are "online services with features that enable members to communicate with each other"[11]; they have an underlying premise that peers offer meaningful support due to the shared experience of a particular life event[12]. They may be created, and moderated, by peers (i.e., those with a direct lived experience of the condition), caregivers, charities, or health professionals. The growing need for online support groups is showcased by the large membership of many groups. For example, at the time of writing a diabetes Facebook group has reached 102,000 members in four years, with 202 new members in the last week, and 202 posts per month[13], and a Long Covid support group has reached 66,000 members, with 96 members in the last week and 2000 posts in the last month[14]. An advantage of these online support groups, as opposed to in-person groups, is that they can transcend geographical boundaries and are less restricted by time or location, which is particularly beneficial to those with limited mobility and those living in rural communities[15]. Such groups can be synchronous via audio or video calls, or they can be asynchronous via social media platforms, such as Facebook groups and discussion boards, or via direct messages, such as in WhatsApp groups[15].

Previous reviews, exploring experiences of online support groups for specific chronic conditions, such as HIV[16] and cancer[17], report that they are a place where group members can receive social support and experience a sense of community, which can result in increased adaptive coping and reduced loneliness. However, they also report that group content can be negative (e.g., distressing personal information or complaints), and that lack of replies and absence of nonverbal communication can lead to misunderstandings and distress. Previous reviews have also explored online support groups for multiple chronic conditions, including how online groups influence daily life[18] and illness self-management[19]. However, these reviews excluded quantitative studies, such as intervention studies, which could provide strong evidence for the impact of online support groups on group member experiences. A meta-analysis exploring health outcomes in relation to online support groups for health conditions did include intervention studies, but only those with a fixed start and end point and included an educational component, which is not representative of existing online support groups[20]. The outcomes included were also limited to social support, depression, quality of life and self-efficacy. Thus, there is a gap in the literature regarding a systematic review on the effects on health and wellbeing of using an online support group which includes both qualitative and quantitative studies and covers a greater number of health and wellbeing outcomes.

In addition to understanding the health impacts of online support groups, it is also important to consider how these effects occur. Previous reviews highlight the importance of finding and exchanging information, receiving emotional support, and sharing experiences[21–24]. Furthermore, the SCENA Model of Therapeutic Affordances of Social Media[25] has also been applied to online support groups[26], and suggests that such groups may afford self-presentation (*managing how one presents themselves online*), connection (*connecting with, and supporting, others*), exploration (*seeking information and improving knowledge*), narration (*exchanging experiences)* and adaptation (*adapting self-management needs in relation to health status*). Due to the variety of platforms used for online support groups (e.g., video- or text-based), as well as the different ways of engaging with the groups (i.e., being a passive or active member of the group), it is important to consider how these factors also influence the health benefits afforded by online support groups[27]. For example, exploration may be easier in text-based groups where there is an archive of information. Previous studies have also explored the role of engagement[28] and group features (e.g., group size, duration, nature of communication)[20]. For example, more social support was reported when online support groups were of a longer duration and included both synchronous and asynchronous channels[20]. Fewer studies have compared asynchronous and synchronous platforms. Furthermore, there is not a review looking at the potential mechanisms underlying each type of health outcome and synthesising group and usage characteristics as

well as support group content in the context of online support groups for chronic conditions.

## Current study
As the number of individuals experiencing, and having their lives disrupted by, chronic conditions increase, it is important to explore potential ways to improve health outcomes. One such way is online support groups. Therefore, it is important to understand the impact of these groups on the health and wellbeing of group members and to identify any influencing factors. This systematic review aims to explore this with the following research questions:

1. What are the effects of online support groups on the observed and self-reported health and wellbeing of individuals with a chronic condition?
2. What are the mechanisms by which online support groups affect the health and wellbeing of individuals with a chronic condition?

## Methods
### Protocol and registration
This systematic review was conducted in concordance with the Preferred Reporting Items for Systematic Reviews and Meta-Analyses (PRISMA) guidelines (Supplementary Table 1)[29]. The systematic review was pre-registered prior to the search with Prospero, registration number: CRD42023399258.

The final review deviated from the pre-registration protocol reported, as the authors did not repeat the search for conditions that were not in the initial search strategy (e.g., endometriosis). The number of research questions differ from the pre-registration. We are no longer comparing outcomes between different types of chronic conditions due to the small number of papers for most of the included chronic conditions.

### Search criteria
In line with recommendations[30], the following databases were searched for peer-reviewed publications on September 11th, 2024:

- Embase 1974 to September 11, 2024
- Ovid MEDLINE® ALL 1946 to September 11, 2024
- APA PsycInfo 1806 to September Week 1 2024
- Web of Science Core Collection

Grey literature searches were also conducted to identify any eligible reports not published via academic publishers, to ensure comprehensiveness, on November 29th 2024, using Google Advanced Search (first 200 items), and Google Scholar (first 200 items). The British Library directory of online doctoral theses (EThOS) was searched on February 14th 2023, without any date restrictions. MedRxiv and PsyArXiv were searched to identify any pre-publication articles uploaded between January 1st, 2023 and September 11th 2024, where we searched the 200 most relevant articles.

Search terms were based on the target population (i.e., those with a chronic condition) and intervention (i.e., online support groups). To avoid unintentionally excluding articles, the study outcomes were not included in the search terms as they relate more broadly to health and wellbeing as opposed to specific outcomes (e.g., depression). Search terms were developed by the research team based on previous reviews on similar topics[16,31], the types of chronic conditions listed by the National Health Service (NHS)[1] and preliminary literature searches. See Supplementary Tables 2–6 for the full search strategy.

### Eligibility criteria
The full inclusion and exclusion criteria are detailed in Table 1. The review included quantitative, qualitative, and mixed method studies (excluding reviews, conference abstracts and protocols). Studies from any country were included, if they were published in English, due to the languages spoken by the research team. The review used the following definition of chronic conditions when deciding eligibility of studies: a health problem that requires ongoing management over a period of years or decades and is one that cannot currently be cured, but can be controlled with the use of medication and/or other therapies[1].

**Table 1 | Inclusion and exclusion criteria**

| Area of study | Inclusion criteria | Exclusion criteria |
|---|---|---|
| Population | Individuals experiencing symptoms of, or diagnosed with, a chronic condition | Individuals not experiencing symptoms of, or not diagnosed with, a chronic condition. |
| Intervention | All types of online peer support groups, involving more than two participants, will be included. The support groups can be asynchronous (e.g., social media group, forum, email) or synchronous (e.g., video call) and can be text- and/or video-based. The review will also include support groups with or without a moderator (led by either a peer (i.e., someone with lived experience) or professional). | In-person support groups, unless they are a comparator for online support groups, will be excluded. The study will also exclude complex interventions (i.e., when a support group is one component of a broader intervention), unless the study specifically reports the independent effects of online support groups. |
| Comparator | The review will include studies with or without a control group. Control groups may include, but are not limited to, in-person support groups, waitlist or standard care. | N/A |
| Outcome | Changes in physical health may include but are not limited to: symptom presence; symptom duration; symptom severity; and changes in limitations in activities due to such physical symptoms[144,145]. Changes in wellbeing may include, but are not limited to: psychological functioning, social relationships, meaning making and changes in limitations in activities due to such symptoms[144,145]. These can be clinical or self-report measures. | Outcomes that are not related to the health and wellbeing of support group members. For example, studies assessing the acceptability of support groups. |

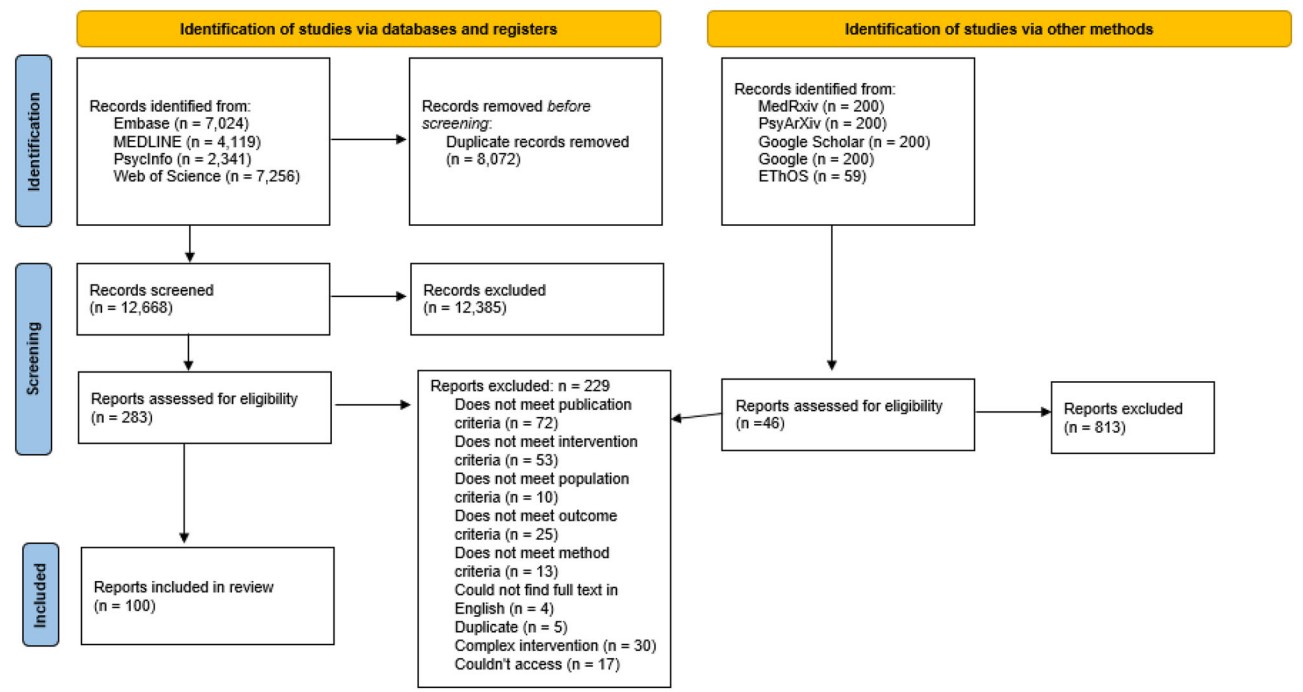

**Fig. 1 | PRISMA flow diagram of the identification of studies.**

## Study selection

Results of the literature searches were exported into the review screening website Rayyan[32]. The first author conducted initial title and abstract screening, where each title was categorised to be either 'included' or 'excluded' for full-text screening. To improve the robustness of the review process 20% of articles underwent title and abstract screening by the third author[33]. The authors agreed on 96% of the articles.

The 'include' articles then underwent full text screening by the first author whereby all articles were categorised into either 'include' or 'exclude'. During full-text screening, the inclusion criteria were tightened to ensure the studies specifically refer to online support group use; to focus on individuals currently experiencing the chronic condition; to exclude studies that focused exclusively on chronic mental, as opposed to chronic physical, health conditions; and to exclude mechanisms relating to offline influences. The third author screened all of the excluded studies[33], where there was 100%

agreement. The screening process can be seen in the PRISMA flowchart in Fig. 1.

## Data extraction and synthesis

Data were extracted in a tabular form on Microsoft Excel by the first author. Results were synthesised using a data-based convergent approach (also called an integrated approach), whereby quantitative and qualitative studies are analysed using the same synthesis method and results are presented together[34,35]. Quantitative data underwent data transformation, which involved creating textual descriptions of quantitative findings. Findings of health and wellbeing were coded in themes using thematic synthesis. Each finding was first coded as 'outcome' or 'mechanism'. To organise the data, each health outcome was coded into themes (e.g., physical health). Findings within each theme were then coded into sub-themes based on the specific finding (e.g., pain), with similar codes being grouped together. This was an iterative process with the grouping of codes and themes changing following

discussions amongst the research team. For mechanisms, each finding was reported in relation to their respective health outcome and were thematically grouped for the discussion (e.g., support). A thematic synthesis was deemed appropriate for the research questions, rather than a meta-analysis, due to the heterogeneity of quantitative studies across health outcomes, in terms of research design and findings. Indeed, whilst all included studies explore the role of online support groups on group members' health and wellbeing, some focus on specific aspects of the groups (e.g., receiving information or level of engagement), whilst others compare the groups to control groups, such as receiving education or treatment as usual.

## Quality assessment
The Mixed Methods Appraisal tool (version 2018) was used to evaluate the quality of included studies[36]. This was a suitable appraisal tool as it was designed for systematic reviews that include qualitative, quantitative, and mixed methods studies. The tool comprises of two questions that apply to all studies, followed by five questions relevant to each methodology. The first author carried out the quality appraisal.

## Reporting summary
Further information on research design is available in the Nature Portfolio Reporting Summary linked to this article.

## Results
### Study selection
In total, 21,599 results were extracted from electronic databases and grey literature searches. Duplication screening was conducted on Rayyan, resulting in 13,527 articles for title and abstract screening. Full text screening was conducted on 329 papers, with 100 papers being included in the final review.

### Study characteristics
A summary of the characteristics of each study can be found in Table 2. Numbers of articles excluded for each reason during the full text screening can be found in Supplementary Table 7. Across the 100 included papers, 24 chronic conditions were included, with breast cancer ($n = 21$), other types of cancer ($n = 18$), HIV/AIDS ($n = 9$) and diabetes ($n = 7$) being the most frequently studied. A full list of included chronic conditions can be found in Table 3.

Most studies were conducted by authors based in the USA ($n = 46$), followed by the UK ($n = 18$), and the Netherlands ($n = 7$). 30 studies did not report participant location. Amongst those that did, most participants were based in the USA ($n = 18$), the Netherlands ($n = 7$), the UK ($n = 9$) and China ($n = 5$) or were international, but with a high proportion of participants in the USA ($n = 7$). These numbers were identified based on inclusion criteria or recruitment details (e.g., recruited via a specific hospital, university, or a location-specific support group). Studies were published between 2002 and 2024. The years with the largest number of published studies were 2021 ($n = 11$), 2022 ($n = 10$), 2023 ($n = 8$) and 2024 ($n = 8$). Most studies recruited participants from either online support groups or through hospitals. Sample sizes of the included studies ranged from 6 to 1641 participants. The effects of online support groups were tested with a variety of methods with the most frequent being cross-sectional quantitative surveys ($n = 43$), cross-sectional interviews ($n = 26$) and quasi-experimental studies ($n = 22$). Experimental studies introduced participants to a new online support group, often created for the experiment whereas cross-sectional studies and longitudinal surveys were naturalistic as they typically assessed the impact of groups in which participants were already a member. Interventions lasted between 1 and 6 months, whilst the duration of support group membership, in cross-sectional studies, when reported, ranged between less than 1 week to 15 years with reported mean duration being between 12 and 31 months. Groups created for the purpose of the research were mostly moderated by the researchers, psychologists, healthcare professionals or patient organisations, whereas studies exploring naturalistic groups often did not report how the group was moderated. 70 papers looked at asynchronous groups (e.g., discussion forums, Facebook groups, WhatsApp groups, email lists), eight papers (of which seven were experimental) looked at synchronous groups (e.g., real time text-based chat groups, or video or teleconference calls), and two explored a combination of both synchronous and asynchronous groups.

### Quality appraisal
The MMAT checklist can be found in Supplementary Tables 8–12. The authors of the MMAT recommend against calculating an overall score for each study, as it is not informative, and instead suggest describing the overall quality of the studies included within the review[36]. Overall, the quality of the studies was satisfactory. Most quantitative descriptive and qualitative papers used opportunistic sampling as participants were recruited via adverts posts in online support groups, so it was often not possible to identify non-response bias. Many authors acknowledged use of a self-selected sample, with participants potentially differing those who did not take part. Similarly, most studies did not discuss characteristics of the target population, so it is not possible to identify whether the samples are representative of other individuals with the chronic condition or representative of online support group members. Furthermore, many studies used standardised scales and statistical analyses, but these often differed for each health outcome thus making it difficult to compare across studies. For example, at least eight scales were used to measure self-efficacy. Furthermore, some papers only reported percentage agreements to health and wellbeing-related statements. Randomised and non-randomised (including longitudinal intervention and naturalistic studies) typically used standardised measures and accounted for confounders in their analysis (e.g., demographics or baseline scores). When participants did leave the study, some studies reported their reasons and statistical differences in baseline scores, but not all.

### Synthesis
The sections below present findings in relation to six health and wellbeing outcomes: physical health, mental health, quality of life, social wellbeing, behaviour and decision-making, and adjustment. Broad mental health and wellbeing was the most frequently explored outcome ($n = 45$), followed by self-efficacy ($n = 22$), and depression and loneliness and isolation ($n = 21$). Table 4 details the types of research method used for each outcome. Tables 5–7 present findings on how usage characteristics and group type may influence each health outcome and provide a summary of the findings.

### Physical health
Physical health outcomes included symptoms and functioning, and pain.

#### Symptoms and functioning
Outcome. The two RCTs found no effect of online support groups on symptoms and functioning over time[37] or compared to website controls[37,38]. Similarly, longitudinal surveys found no effect of joining an asynchronous group on their health status[39] nor any differences between users and non-users on functional wellbeing[40]. However, in post-intervention interviews following a non-randomised control trial, 53% of participants reported that participating in an online support group contributed to a reduction in their symptoms[41] and 86% of (seven) participants agreed that the posts in a secret Facebook page were helpful in improving their recovery[42]. Furthermore, one cross-sectional survey found lower self-reported symptom scores and higher function scores in online support group members compared to members of a face-to-face support group[43]. A cross-sectional analysis of health records found that patients with diabetes from a closed Facebook group had lower blood sugar levels compared to those not in the Facebook group, but there were no differences in other health outcomes[44]. Additionally, two cross-sectional qualitative studies, reported improved symptoms, enhanced functional wellbeing and expedited recovery[45,46], although this was not the case for all group members[46]

Mechanisms. Two cross-sectional surveys found that online social, emotional, and informational support was positively related to physical quality

**Table 2 | Included studies**

| First author | First author location | Study design | Chronic Condition | Online support group description | Comparator description | Health and Wellbeing Outcome |
|---|---|---|---|---|---|---|
| Algtwei et al. (2017)[146] | UK | Cross-sectional survey (n = 199) | Cancer | Online supports with at least 50 members and at least 25 threads within the past 30 days | N/A | Depression<br>Anxiety<br>Quality of life<br>Self-efficacy |
| Ashtari and Taylor (2022)[58] | USA | Cross-sectional interview (n = 30) | Ehlers-Danlos Syndrome | Online peer support groups hosted by Facebook, Instagram, Twitter, Reddit and Inspire | Not participating in an online support group to find information (20%) | Pain<br>Broad mental health and wellbeing<br>Behaviour change<br>Treatment decision-making<br>Feeling less alone<br>Identity |
| Ashtari and Taylor (2023)[66] | USA | Mixed methods: Cross-sectional survey (n = 413) Cross-sectional interview (n = 30) | Ehlers-Danlos Syndrome | Not specified | N/A | Broad mental health and wellbeing<br>Feelings of belonging |
| Babyar (2016)[115] | USA | Cross sectional survey (n = 57) | Cystic Fibrosis | Not specified | N/A | Adherence |
| Bartlett and Coulson (2011)[102] | UK | Cross-sectional survey (n = 246) | Various conditions | Active online support groups with more than 5 members with the primary aim of providing support for patients with a chronic physical illness | N/A | Broad social wellbeing<br>Treatment decision-making<br>Illness acceptance<br>Self-esteem<br>Optimism and control |
| Batenburg and Das (2014)[61] | The Netherlands | Longitudinal survey (n = 133) | Breast cancer | Not specified | N/A | Broad mental health and wellbeing<br>Depression |
| Batenburg and Das (2014)[62] | The Netherlands | Cross-sectional survey (n = 175) | Breast cancer | Dutch peer-led message boards available 24/7 with messages posted in the last month | N/A | Broad mental health and wellbeing<br>Depression |
| Batenburg and Das (2015)[86] | The Netherlands | Cross-sectional survey (n = 114) | Breast cancer | Online 24 h-available peer led message boards on breast cancer with messages posted in the last month | N/A | Broad mental health and wellbeing<br>Depression |
| Baydoun et al. (2021)[59] | Canada | RCT (n = 125) | Breast cancer | 90 minute once weekly professionally moderated and guided discussions in a synchronous online text based chat session (plus NuCare) | NuCare: self-guided psychoeducational manual focused on adopting new ways of coping | Broad mental health and wellbeing<br>Depression<br>Anxiety<br>Loneliness and isolation<br>Coping |
| Bazrafshani et al. (2022)[45] | Iran | Mixed methods: Cross-sectional interview (n = 29) Delphi study (n = 26) | HIV/AIDS | Online social networks e.g., Telegram, WhatsApp and Facebook | N/A | Symptoms and functioning<br>Broad mental health and wellbeing<br>Feelings of belonging<br>Loneliness and isolation<br>Behaviour change<br>Motivation<br>Adherence<br>Treatment decision making<br>Self-efficacy<br>Empowerment<br>Optimism and hope<br>Self-esteem<br>Coping |
| Beaudoin and Tao (2007)[95] | USA | Cross-sectional survey (n = 372) | Cancer | Email discussion groups and instant messages | N/A | Depression<br>Coping |
| Brady et al. (2017)[112] | UK | Cross-sectional interview (n = 21) | Diabetes | Not specified | N/A | Behaviour change<br>Motivation<br>Treatment decision-making<br>Self-efficacy<br>Empowerment<br>Identity |
| Changrani et al. (2008)[56] | USA | RCT (n = 68) | Cancer | Virtual Community for Immigrants with Cancer. 8 people in each online group, meeting weekly for 90 minutes and facilitated by trained bilingual facilitators to discuss issues of interest | Usual care | Pain<br>Depression<br>Quality of life<br>Post-traumatic growth |

**Table 2 (continued) | Included studies**

| First author | First author location | Study design | Chronic Condition | Online support group description | Comparator description | Health and Wellbeing Outcome |
|---|---|---|---|---|---|---|
| Chen et al. (2020)[101] | China | Cross-sectional survey ($n = 1241$) | Diabetes | 73% were members of groups with more than 100 members | N/A | Connections and friendships Behaviour change Quality of life Self-efficacy |
| Chung et al. (2021)[99] | USA | Cross-sectional survey ($n = 124$) | Parkinson's Disease (PD) | Run by Parkinson's patients; not commercial operations; active; numbers from 1600 - 2900 members | N/A | Quality of life |
| Cooper et al. (2021)[42] | USA | Quasi-experiment ($n = 9$) | HIV | Secret Facebook group | N/A | Symptoms and functioning Quality of life |
| Costello et al. (2019)[39] | UK | Longitudinal survey ($n = 329$) | Variety of conditions | HealthUnlocked: a host to 700 online health communities built in collaboration with patient organisations, who moderate the communities. Registered members can follow communities, post questions, updates or reply to other posts. They can post text and images | N/A | Symptoms and functioning Self-efficacy |
| Coulson (2013)[72] | UK | Cross-sectional open-ended survey ($n = 249$) | Inflammatory bowel disease (e.g., Crohn's disease or ulcerative colitis) | Asynchronous online support communities (e.g., discussion forums) | N/A | Broad mental health and wellbeing Anxiety Loneliness and isolation Behaviour change Illness acceptance Feeling less alone Feeling understood and reassured Optimism and hope Self-esteem |
| Cummings et al. (2002)[67] | USA | Cross-sectional survey ($n = 64$) | Hearing impairment | Beyond hearing: email distribution list ($n = 240$) for people who have hearing loss. All emails sent to the list are automatically emailed to other group members. | N/A | Broad mental health and wellbeing |
| Day (2022)[73] | UK | Cross-sectional interview (n = 11) | Long Covid | Facebook and Whatsapp groups | N/A | Broad mental health and wellbeing Feeling understood and reassured Optimism and hope |
| Egerton et al. (2022)[87] | Australia | Post-intervention interview ($n = 10$) | Knee osteoarthritis | My Knee Community: expert-moderated online discussion forum organised into categories and threads. Moderators monitor the posts in the group, post information and answer questions if they requested a response from a healthcare professional. Members could add posts to threads or create new threads | Web-based patient education resource including factsheets, videos and other tools (e.g., risk assessments) | Broad mental health and wellbeing Behaviour change Motivation Feeling understood and reassured |
| Fullwood et al. (2019)[124] | UK | Cross-sectional survey ($n = 271$) | Various conditions | Not specified | N/A | Illness acceptance |
| Garrett et al. (2024)[76] | USA | Cross-sectional interview ($n = 15$) | Long Covid | Social media, specifically Facebook groups | N/A | Broad mental health and wellbeing Anxiety Connections and friendship Loneliness and isolation Feeling less alone Feeling understood and validated |
| Han et al. (2014)[40] | USA | Quasi-experiment ($n = 325$) | Breast cancer | Discussion forum component of CHESS: a text-based, asynchronous bulletin board monitored by a trained facilitator | N/A | Symptoms and function Depression Connections and friendship Self-efficacy |
| Han et al. (2019)[94] | USA | Quasi-experiment ($n = 236$) | Breast cancer | Discussion forum component of CHESS: a text-based, asynchronous bulletin board monitored by a trained facilitator | N/A | Depression Coping |
| Hansen et al. (2021)[111] | Norway | Cross-sectional survey ($n = 540$) | Type 2 Diabetes | Not specified | Not participating in an online support group | Behaviour change |

**Table 2 (continued) | Included studies**

| First author | First author location | Study design | Chronic Condition | Online support group description | Comparator description | Health and Wellbeing Outcome |
|---|---|---|---|---|---|---|
| Healy et al. (2022)[125] | USA | Cross-sectional interview (n = 35) | HIV | Existing, organically developed youth WhatsApp groups at HIV care clinics | Not part of a WhatsApp group | Feeling less alone |
| Herrero et al. (2019)[63] | Spain | Cross-sectional survey (n = 307) | Diabetes | Facebook group, diabetes-related forum, health related forum and other types of online forums | Not belonging to an online support group (39.1% of the sample) | Broad mental health and wellbeing Motivation Self-esteem |
| Herrero et al. (2021)[110] | Spain | Cross-sectional survey (n = 307) | Diabetes | Facebook groups Diabetes related forums Health related forums | Not belonging to an online support group (32% of the sample) | Behaviour change |
| Hodson and O'Meara (2023)[93] | Canada | Cross-sectional interview (n = 12) | Cancer | Social media, specifically Facebook groups | N/A | Depression Anxiety |
| Holbrey and Coulson (2013)[77] | UK | Cross-sectional open-ended survey (n = 55) | Polycystic Ovary Syndrome (PCOS) | A volunteer run, peer-led discussion forum with 4,600 members. It has been identified as one of the most active for women suffering with PCOS | N/A | Broad mental health and wellbeing Anxiety Feelings of belonging Treatment decision-making Self-efficacy Optimism and hope |
| Holdren et al. (2023)[105] | USA | Mixed methods: Cross-sectional interview (n = 20) Cross-sectional survey (n = 291) | Cancer | Closed Facebook cancer peer support groups | N/A | Connections and friendship Feeling less alone |
| Huber et al. (2017)[43] | Germany | Cross-sectional survey (n = 1641) | Cancer | Largest German online support group with 90,000 postings from 4,400 registered users | Face to face support group with a group leader that organises meetings and invites guest speakers | Symptoms and functioning Broad mental health and wellbeing Depression Anxiety Distress Quality of life Treatment decision-making Illness acceptance Feeling less alone Self-esteem |
| Huber et al. (2017)[119] | Germany | Cross-sectional survey (n = 686) | Cancer | German prostate cancer online group (n = 3357) | N/A | Treatment decision making |
| Hurtado Illanes (2024)[68] | Spain | Cross-sectional qualitative survey (n = 26) | Chronic patient | Virtual community support | Blogs | Broad mental health and wellbeing Feelings of belonging Behaviour change Treatment decision-making Self-efficacy |
| Iliffe and Thomspon (2019)[74] | UK | Cross-sectional interview (n = 12) | Alopecia | Alopecia UK Facebook peer support group | N/A | Broad mental health and wellbeing Feelings of belonging Loneliness and isolation Treatment decision-making Illness acceptance Feeling less alone Identity Coping |
| Kaal et al. (2018)[106] | The Netherlands | Cross-sectional survey (n = 30) | Cancer | An online community with a community manager (non-health care professional) who is online for 4 hours per day. | N/A | Connections and friendship Loneliness and isolation Feeling understood and reassured Self-esteem |
| Kever et al. (2022)[41] | USA | Non-randomised control trial: Efficacy outcomes (n = 31) Focus group evaluation (n = 17) | Multiple Sclerosis (MS) | Participants met in an online group of 10 individuals led by a psychologist where group members were encouraged to share challenges, fears, thoughts and coping strategies. Each meeting starts with a 'win'. Discussions evolved based on participants' current needs and the input they provided. | No treatment control (10 participants) | Symptoms and functioning Broad mental health and wellbeing Depression Anxiety Distress Quality of life Loneliness Self-efficacy Feelings of belonging |
| Kim et al. (2012)[85] | USA | Quasi-experiment (n = 177) | Breast cancer | Discussion forum component of CHESS: a text-based, asynchronous bulletin board monitored by a trained facilitator | N/A | Broad mental health and wellbeing Illness acceptance |

**Table 2 (continued) | Included studies**

| First author | First author location | Study design | Chronic Condition | Online support group description | Comparator description | Health and Wellbeing Outcome |
|---|---|---|---|---|---|---|
| Kim et al. (2017)[100] | USA | Quasi-experiment (n = 221) | Breast cancer | Discussion forum component of CHESS: a text-based, asynchronous bulletin board monitored by a trained facilitator | N/A | Quality of life<br>Coping |
| Klemm (2012)[88] | USA | RCT (n = 50) | Breast cancer | Asynchronous online support group moderated by social workers. Moderators introduced weekly topics and facilitated discussions among group members. | Asynchronous peer-led online support groups, with open discussion without the use of preselected topics or input from a moderator | Depression |
| Kosugi et al. (2021)[147] | Japan | Cross-sectional survey (n = 334) | Cancer | Cancer Parents: one of the largest online peer support groups in Japan for adult cancer patients (n = 3000) | N/A | Loneliness and isolation |
| Koufopoulos et al. (2016)[114] | UK | RCT (n = 216) | Asthma | Asthma Village: an online community for patients with asthma where participants could leave comments or see who else was online. There was no experimenter intervention | Asthma Diary: an online diary for recording ICS preventer use. Participants could not read the posts of others or interact with others | Treatment adherence |
| Lange et al. (2017)[60] | Germany | Quasi-experiment (n = 44) | Cancer | Internet chat programme where prostate cancer patients met for 60-90 minutes. Each session was guided by psychotherapists and had a topic focus | Treatment as usual | Broad mental health and wellbeing<br>Depression<br>Anxiety<br>Quality of life<br>Coping |
| Leavitt et al. (2020)[90] | USA | Non-randomised controlled trial survey (n = 28) | Multiple Sclerosis (MS) | eSupport: private facilitated video calls with 6 members per group | eJournal: weekly online semi-structured journalling activities | Depression<br>Loneliness |
| Letourneau et al. (2012)[104] | Canada | Pre/Post quantitative survey (n = 10) Post-intervention qualitative survey and interview (n = 14) | Asthma and Allergies | Weekly synchronous chat sessions centred around specific topics; and an internet community bulletin board; and emails | N/A | Connections and friendships<br>Loneliness and isolation<br>Self-efficacy<br>Coping |
| Lieberman et al. (2003)[57] | USA | Pre/Post survey (n = 32 adult) | Breast carcinoma | Electronic support group: 8 people per group meeting for 1.5 hours once a week with no set agenda. Also had access to a private newsgroup where they could chat, post pictures and share their stories | N/A | Pain<br>Depression<br>Anxiety<br>Post-traumatic growth<br>Coping |
| Lieberman (2007)[50] | USA | Longitudinal survey (n = 77) | Breast cancer | Breast cancer bulletin boards that had more than 20 postings a day | N/A | Symptoms and functioning<br>Depression<br>Quality of life |
| Lieberman and Winzelberg (2009)[53] | USA | Pre/Post survey (n = 91) | Breast cancer | 5 online peer led asynchronous groups with large active membership with women usually responding within a day to new members. There's a core of active members that provide continuity and share views on how to cope with the disease and how to fight and actively combat the disease. | N/A | Symptoms and functioning<br>Broad mental health and wellbeing<br>Depression<br>Broad social wellbeing |
| Litchman (2015)[49] | USA | Cross-sectional survey (n = 183 adults) Cross-sectional interview (n = 20) Cross-sectional survey (n = 178) | Diabetes | 4 online communities:<br>1) TuDiabetes: sharing of blog posts, discussions, events, photos and videos on over 22,000 forums and 400 groups<br>2) Diabetic Connect: discussion boards and options to share and rate receipes and medication and updated new stories and Ask an Expert<br>3) Facebook<br>4) Twitter | N/A | Symptoms and functioning<br>Broad mental health and wellbeing<br>Quality of life<br>Connections and friendships<br>Feelings of belonging<br>Behaviour change<br>Treatment decision-making<br>Self-efficacy<br>Empowerment<br>Feeling less alone<br>Feeling understood and reassured |

**Table 2 (continued) | Included studies**

| First author | First author location | Study design | Chronic Condition | Online support group description | Comparator description | Health and Wellbeing Outcome |
|---|---|---|---|---|---|---|
| Lopez-Olivo et al. (2022)[38] | USA | RCT (*n* = 204) | Rheumatoid Arthritis | Secret Facebook community and access to an educational website. Online community was peer led by 2 advocates / spokespeople for rheumatoid arthritis. Different topics introduced each week for discussion | Educational website | Symptoms and functioning<br>Pain<br>Broad mental health and wellbeing<br>Depression<br>Anxiety<br>Quality of life<br>Behaviour change<br>Self-efficacy<br>Empowerment |
| Mackie et al. (2024)[107] | Australia | Cross-sectional interview (*n* = 28) | Breast cancer | Teleconference | N/A | Connections and friendship<br>Empowerment:<br>Optimism and hope<br>Self-esteem |
| Meade et al. (2013)[108] | UK | Cross-sectional interview (*n* = 6) | Neuromuscular disorder | Emails to previous participants and general advertisement on the message board | N/A | Connections and friendship<br>Loneliness and isolation<br>Feeling less alone<br>Feeling understood and reassured<br>Coping |
| Meng et al. (2021)[84] | USA | Cross-sectional survey (*n* = 386) | Cancer | Two online communities providing basic social networking features (e.g., personal profile / adding friends) and discussion forums where cancer patients can exchange social support | N/A | Broad mental health and wellbeing |
| Mills et al. (2024)[46] | UK | Cross-sectional interview (*n* = 21) | Long Covid | Facebook, WhatsApp and Zoom groups. | N/A | Symptoms and functioning<br>Broad mental health and wellbeing<br>Feelings of belonging<br>Connections and friendship<br>Loneliness and isolation<br>Self-efficacy<br>Feeling less alone |
| Mo and Coulson (2010)[28] | UK | Cross-sectional survey (*n* = 640) | HIV/AIDS | Using an online support group | Not using an online support group | Symptoms and functioning<br>Pain<br>Broad mental health and wellbeing<br>Broad social wellbeing<br>Coping |
| Mo and Coulson (2010)[148] | UK | Cross-sectional survey (*n* = 340) | HIV/AIDS | Online support groups with at least 25 message threads within the last 30 days and at least 50 members | N/A | Symptoms and functioning<br>Pain<br>Broad mental health and wellbeing<br>Depression<br>Distress<br>Quality of life<br>Broad social wellbeing<br>Loneliness and isolation<br>Self-efficacy<br>Optimism and hope<br>Coping |
| Mo and Coulson (2012)[122] | Hong Kong | Cross-sectional survey (*n* = 340) | HIV/AIDS | Not specified | N/A | Self-efficacy<br>Coping |
| Mo and Coulson (2013)[123] | Hong Kong | Cross-sectional survey (*n* = 340) | HIV/AIDS | Public groups with at least 10 new messages posted to the group each day | N/A | Depression<br>Loneliness and isolation<br>Illness acceptance<br>Optimism and hope |
| Mo and Coulson (2014)[78] | Hong Kong | Cross sectional open-ended survey (*n* = 115) | HIV/AIDS | Not specified | N/A | Broad mental health and wellbeing<br>Anxiety<br>Connections and friendships<br>Loneliness and isolation<br>Behaviour change<br>Feeling understood and reassured<br>Feeling less alone<br>Optimism and hope |
| Morehouse et al. (2021)[69] | USA | Cross-sectional quantitative and qualitative survey (*n* = 76) | Myalgic encephalomyelitis /chronic fatigue syndrome (ME/CFS) | 18 online support groups with a total of 2000 members | N/A | Broad mental health and wellbeing<br>Feelings of belonging<br>Loneliness and isolation<br>Behaviour change<br>Feeling understood and reassured<br>Self-esteem |

**Table 2 (continued) | Included studies**

| First author | First author location | Study design | Chronic Condition | Online support group description | Comparator description | Health and Wellbeing Outcome |
|---|---|---|---|---|---|---|
| | | | | | | Optimism and hope<br>Coping |
| Parrish (2011)[70] | USA | Cross-sectional survey (n = 472) | Cancer | Private Listserv group and its subgroup | N/A | Broad mental health and wellbeing<br>Distress<br>Quality of life<br>Connections and friendship<br>Feeling less alone<br>Identity |
| Parrocha and Bernadas (2024)[126] | Philippines | Cross-sectional interview (n = 15) | Cardiovascular disease | Private Facebook group with 3500 members | N/A | Feeling less alone |
| Pester et al. (2022)[55] | USA | Randomised controlled trial n = 119 | Chronic pain | Professional-led Facebook groups: investigators posted nearly every morning to disseminate training components | Standard Facebook groups: encouraged by the investigators to offer mutual support for the duration of the group | Pain<br>Depression<br>Anxiety |
| Petrovski and Zivkovic (2019)[44] | Macedonia | Cross-sectional analysis of electronic health records (n = 728) | Diabetes | Internet group: standard medical protocol plus active members of national closed Facebook group on diabetes | Non-internet group: standard medical protocol with regular clinic visits | Symptoms and functioning |
| Ronen et al. (2023)[89] | USA | Pre/Post quantitative survey (n = 55) | HIV | Interactive WhatsApp groups (n = 25) facilitated by study staff for 6 months who sent weekly structured messages. | N/A | Depression<br>Behaviour<br>Motivation<br>Adherence<br>Self-efficacy |
| Rose et al. (2024)[120] | USA | Cross sectional survey (n = 219) | Cancer | 5 Facebook groups for scalp cooling (n = 247 - 9047 members) | N/A | Feelings of belonging<br>Treatment decision-making |
| Rowlands et al. (2023)[79] | UK | Cross-sectional focus groups (n = 10) | Various conditions | Not specified | N/A | Broad mental health and wellbeing<br>Feeling understood and reassured<br>Optimism and hope |
| Russell et al. (2022)[98] | USA | Cross-sectional interview (n = 20) | Long Covid | Long Covid online community including Reddit forums, Facebook groups and Slack channels | N/A | Anxiety<br>Empowerment<br>Illness acceptance |
| Salzer et al. (2010)[37] | USA | RCT (n = 78) | Breast cancer | Unmoderated closed Listserv (email list) | Review of information on a cancer-related website | Symptoms and functioning<br>Broad mental health and Wellbeing<br>Distress<br>Self-efficacy<br>Optimism and hope |
| Seçkin (2007)[71] | USA | Cross-sectional survey (n = 350) | Cancer | Active online support groups (i.e., new messages posted on a daily basis) | N/A | Broad mental health and wellbeing<br>Self-efficacy<br>Illness acceptance<br>Optimism and hope<br>Coping |
| Seçkin (2011)[127] | USA | Cross-sectional survey (n = 255) | Cancer | Not specified | N/A | Optimism and hope |
| Setoyama et al. (2011)[96] | Japan | Cross-sectional survey (n = 220) | Breast cancer | Online communities that include only those with breast cancer and are not moderated by healthcare providers. | N/A | Depression<br>Anxiety |
| Shaw et al. (2006)[54] | USA | Quasi-experiment (n = 144) | Breast cancer | Discussion forum component of CHESS: a text-based, asynchronous bulletin board monitored by a trained facilitator | N/A | Symptoms and functioning<br>Broad mental health and wellbeing |
| Shaw et al. (2007)[51] | USA | Quasi-experiment (n = 97) | Breast cancer | Discussion forum component of CHESS: a text-based, asynchronous bulletin board monitored by a trained facilitator | N/A | Symptoms and functioning<br>Broad mental health and wellbeing<br>Self-efficacy<br>Illness acceptance |
| Shaw et al. (2008)[83] | USA | Quasi-experiment (n = 231) | Breast cancer | Discussion forum component of CHESS: a text-based, asynchronous bulletin board monitored by a trained facilitator | N/A | Broad mental health and wellbeing |

**Table 2 (continued) | Included studies**

| First author | First author location | Study design | Chronic Condition | Online support group description | Comparator description | Health and Wellbeing Outcome |
|---|---|---|---|---|---|---|
| Shim et al. (2011)[52] | USA | Quasi-experiment (*n* = 106) | Breast cancer | Discussion forum component of CHESS: a text-based, asynchronous bulletin board monitored by a trained facilitator | N/A | Symptoms and functioning Broad mental health and wellbeing Self-efficacy |
| Shoebotham and Coulson (2016)[26] | UK | Cross sectional qualitative survey (*n* = 69) | Endometriosis | Not specified | N/A | Pain Broad mental health and wellbeing Distress Loneliness and isolation Treatment decision-making Self-efficacy Empowerment Illness acceptance Feeling understood and reassured Coping |
| Sparling et al. (2017)[64] | USA | Cross-sectional survey (*n* = 508) | Multiple Sclerosis | Not specified | N/A | Broad mental health and wellbeing |
| Steadman and Pretorius (2014)[80] | South Africa | Cross-sectional interview (*n* = 10) | Multiple Sclerosis (MS) | Online Facebook support group for people with MS | N/A | Broad mental health and wellbeing Feelings of belonging Connections and friendship Loneliness and isolation |
| Stewart et al. (2013)[103] | Canada | Post intervention interview (*n* = 27) Pre/Post survey (*n* = 27) | Asthma and Allergies | Online weekly video 'real-time' support (45–120 minutes) led by peer mentors and health professionals. Groups were matched by condition and gender. | N/A | Feelings of belonging Loneliness and isolation Behaviour change Self-efficacy Feeling less alone Coping |
| Tam et al. (2023)[117] | USA | Cross-sectional survey (*n* = 97) | Cancer | Facebook groups for head and neck cancer (*n* = 967–6,096) | N/A | Treatment decision-making: |
| Tankha (2023)[149] | USA | RCT (*n* = 119) | Any type of chronic pain | Professional-led Facebook groups: investigators posted nearly every morning to disseminate training components | Standard Facebook groups: encouraged by the investigators to offer mutual support for the duration of the group | Self-efficacy |
| Terborg (2023)[150] | UK | Cross-sectional survey (*n* = 90) | Tinnitus | Posted on Tinnitus Talk | Not using Tinnitus Talk | Behaviour change |
| Thewlis (2021)[65] | Ireland | Cross-sectional survey (*n* = 76) | Chronic illness | Not specified | Not in an online support group | Broad mental health and wellbeing |
| van Uden-Kraan et al. (2008)[22] | The Netherlands | Cross-sectional interview (*n* = 32) | Various conditions | Active online support groups (i.e., 50 posts a month) | N/A | Broad mental health and wellbeing Feelings of belonging Connections and friendships Loneliness and isolation Behaviour change Treatment decision-making Self-efficacy Illness acceptance Feeling less alone Feeling understood and reassured Optimism and hope Self-esteem Coping |
| van Uden-Kraan et al. (2008)[151] | The Netherlands | Cross-sectional survey (*n* = 528) | Various conditions | Posters on Dutch online groups | Lurkers on the same group (defined as never having posted) | Social wellbeing Illness acceptance Self-esteem Optimism and hope |
| van Uden-Kraan et al. (2009)[23] | The Netherlands | Cross-sectional survey (*n* = 528) | Various conditions | Active online support groups (i.e., 30 posts a month) | N/A | Broad social wellbeing Connections and friendships Loneliness and isolation Treatment decision-making Illness acceptance Optimism and hope Self-esteem |
| Vanstrum et al. (2022)[91] | USA | Cross-sectional survey (*n* = 549) | Vestibular disorders | Facebook groups | N/A | Depression Treatment decision-making Optimism and hope Coping |
| Vanstrum et al. (2024)[92] | USA | Cross-sectional survey (*n* = 97) | Vestibular disorders | Online support groups with over 20,000 members for general vestibular symptoms | Face to face support group | Depression Anxiety Treatment decision-making Coping |

**Table 2 (continued) | Included studies**

| First author | First author location | Study design | Chronic Condition | Online support group description | Comparator description | Health and Wellbeing Outcome |
|---|---|---|---|---|---|---|
| Vilhauer (2009)[97] | USA | Post-intervention interview (n = 20) Pre/post survey n = 18 | Breast cancer | Unmoderated email-based support groups. Maximum group membership was limited to 10 or 11 women | Wait list for support group | Anxiety Feelings of belonging Connections and friendships Isolation and loneliness Treatment decision making Empowerment Illness acceptance Optimism and hope |
| Walsh et al. (2024)[81] | USA | Cross-sectional interview (n = 25) | Cancer | 3 specified online support groups | N/A | Broad mental health and wellbeing Connection and friendship |
| Willis et al. (2018)[113] | USA | Cross-sectional interview (n = 20) | Arthritis | Not specified | N/A | Behaviour change Illness acceptance |
| Willis et al. (2018)[116] | USA | Cross-sectional interview (n = 20) | Arthritis | Online health communities dedicated to arthritis and were absent of health care professionals and medical interventions. Each group included discussion boards and archived threads on topics such as medications, exercise and special diet plans | N/A | Treatment adherence Identity |
| Wilson and Stock (2021)[82] | UK | Cross-sectional interview (n = 15) | Long-term conditions (e.g., asthma, chronic fatigue, lupus, fibromyalgia) | Not specified | N/A | Broad mental health and wellbeing Anxiety Feelings of belonging Loneliness and isolation Feeling less alone Identity |
| Wu et al. (2021)[152] | Taiwan | Cross-sectional survey (n = 452) | Chronic patients | Online communities established by hospitals or healthcare professionals | N/A | Adherence |
| Yao et al. (2015)[47] | China | Mixed methods: Cross-sectional interview = (n = not reported) Cross-sectional survey n = 349 | Hepatitis B | The Anti-Heptatitis B Forum - one of the most popular and active | N/A | Symptoms and functioning Broad mental health and wellbeing Anxiety Quality of life Feelings of belonging Connections and friendship Optimism and hope Identity |
| Zheng et al. (2016)[48] | China | Cross-sectional survey (n = 326) | Hepatitis B | The Anti-Heptatitis B Forum - one of the most popular and active | N/A | Symptoms and functioning Broad mental health and wellbeing Quality of life |
| Zhu et al. (2018)[128] | China | Cross-sectional interview (n = 18) | Breast cancer | Discussion forum | N/A | Self-esteem |
| Zigron and Bronstein (2019)[75] | Israel | Cross-sectional interview (n = 23) | Ulcerative colitis and Chron's Disease | Closed Facebook health community with 1875 registered members. Community members can share and consult in these areas, recommend treatments, share experiences | N/A | Broad mental health and wellbeing Feelings of belonging Motivation Empowerment Optimism and hope |

of life[47,48]. Similarly, two interview studies suggested that sharing experiences and information on the group was attributed to improved symptoms and functioning[45,46]. Additionally, in a cross-sectional survey, participants who reported that the online community helped them to learn strategies to improve insurance coverage were more likely to have increased blood sugar levels[49]. However, there were conflicting findings regarding the role of religious expression and insightful disclosure. Of five pre-post content analyses, within intervention and naturalistic settings, three reported that greater religious expression and insightful disclosure by participants were associated with improved self-reported functional wellbeing [50–52], but two did not[53,54]. One also reported no association between disclosure of negative or positive emotions and functional wellbeing[52]. Another cross-sectional survey found no relationship between perceived competence of online discussions and diabetes related complications or blood sugar levels and that participants who reported that the online community helped them to learn

strategies to improve insurance coverage were more likely to have increased blood sugar levels[49].

## Pain

Outcome. One RCT reported a reduction in pain severity and interference amongst members of moderated and unmoderated Facebook support groups[55]. However, two RCTs found no significant change in pain scores between the intervention (weekly moderated synchronous groups and a secret Facebook group plus education) compared to usual care or an educational control[38,56]. A pre-post intervention study reported positive outcomes on reactions to pain[57] and two cross-sectional qualitative studies reported that online support groups helped with pain reduction[26,58]. In particular, a participant reported that suggestions made on online support groups helped them to stay ahead of their pain, when previously they would have gone to hospital[26].

**Table 3 | Chronic Conditions identified in the review**

| Chronic Condition | Number of papers | References |
|---|---|---|
| Breast cancer | 21 | 37,40,50–54,57,59,61,62,83,85,86,88,94,96,97,100,107,128 |
| Other/multiple types of cancer | 18 | 43,56,60,70,71,81,84,93,95,105,106,117–120,127,146,147 |
| Various conditions included | 11 | 22,23,39,65,68,79,82,102,124,151,152 |
| HIV/AIDS | 9 | 28,42,45,78,89,122,123,125,148 |
| Diabetes | 7 | 44,49,63,101,110–112 |
| Arthritis | 4 | 38,87,113,116 |
| Multiple Sclerosis | 4 | 41,64,80,90 |
| Long Covid | 4 | 46,73,76,98 |
| Asthma | 3 | 103,104,114 |
| Chronic pain | 2 | 55,121 |
| Hepatitis B | 2 | 47,48 |
| Inflammatory bowel conditions (Crohn's disease and ulcerative colitis) | 2 | 72,75 |
| Ehlers-Danlos Syndrome | 2 | 58,66 |
| Vestibular disorders | 2 | 91,92 |
| Cystic fibrosis | 1 | 115 |
| Endometriosis | 1 | 26 |
| Hearing impairment | 1 | 67 |
| Myalgic encephalomyelitis/chronic fatigue syndrome (ME/CFS) | 1 | 69 |
| Parkinson's Disease | 1 | 99 |
| Polycystic Ovary Syndrome | 1 | 77 |
| Alopecia | 1 | 74 |
| Cardiovascular disease | 1 | 126 |
| Tinnitus | 1 | 150 |
| Neuromuscular disorders | 1 | 108 |

## Mental health

This section includes broad mental health and wellbeing, depression, anxiety, and distress.

**Broad mental health and wellbeing.** Here we consider measures of emotional benefits, emotional health, (psychological or emotional) wellbeing, negative feelings, difficult emotions, mental health and mood.

Outcome. Three RCTs reported mixed findings. One RCT reported improvements in mood scores across all participants after the intervention and at follow-up, but there were no differences between the intervention group (moderated weekly calls plus education) and the control (education only)[59]. However, another found that no differences in stress scores between a secret Facebook group plus education and an educational control[38], and another RCT found that women in an unmoderated email group had poorer wellbeing at both 4 and 12 months than women using an educational website[37]. Similarly, anger increased over time for participants of a moderated weekly online group and was higher amongst the intervention group than the control group at the end of the study[60]. Moreover, a longitudinal survey reported no change in emotional wellbeing over time[61]. Interviews following a non-randomised controlled trial found that some participants were not as sad as before they joined a weekly professionally moderated group[41]. Furthermore, four cross-sectional quantitative studies also found no association between online support group participation and mental health [62–65], whereas a further six reported a positive effect on mental health and wellbeing [66–71]. For example, in one cross-sectional survey, 100% of participants agreed that the online support group made a positive difference in their emotional health[70] and in another cross-sectional survey, 75% indicated that being involved with online support groups increased their satisfaction with daily life, 57.9% reported reduced sadness, and 27.6% expressed that involvement in the online support group had decreased thoughts of suicide[69]. However, one cross-sectional quantitative survey compared online to face-to-face support groups and found that more people attending a face-to-face group (two-thirds of participants) reported positive wellbeing than those attending an online support group (one-third of participants)[43]. 10 cross-sectional qualitative studies reported a positive effect on mental health and wellbeing[26,45–47,58,66,72–75], but 12 reported reduced wellbeing, including feelings of frustration, fear, upset, sadness, overwhelm, guilt and disappointment[22,46,69,72,73,76–82].

Mechanisms. Content analyses in a pre-post intervention reported that the use of a higher percentage of religious words predicted lower levels of self-reported negative emotions, but not emotional wellbeing[51]. Similarly, insightful disclosure in two intervention studies was predictive of lower levels of self-reported negative emotions and improved emotional wellbeing[52,54]. Furthermore, although a survey found that 85.8% of participants said that writing down thoughts and feelings made them feel better[70], a content analysis in an intervention study found no association between disclosure of negative or positive emotions and emotional wellbeing[52]. Additionally, communicating about oneself within an online support group in an intervention study (measured through first-person pronoun use, e.g., 'I') was associated with higher levels of negative emotions, but communicating about others (measured through use of relational pronouns, e.g., 'we' or 'you') was not[83].

Two cross-sectional quantitative studies found that receiving online emotional support was positively associated with emotional wellbeing[84] and receiving online support was associated with psychological quality of life[48]. However, a further two found that giving and receiving social support, and receiving informational support, was not associated with mental health[84,85]. Online support network size also had an indirect positive effect on emotional wellbeing in a cross-sectional survey, through online received emotional support[84]. Furthermore, a cross-sectional survey found that

**Table 4 | Type and number of study designs measuring each main health outcome and mechanism**

| Health outcome | Total number of papers | Mechanisms only | Randomised controlled trial | Quasi-experimental | Cross-sectional quantitative | Cross-sectional qualitative |
|---|---|---|---|---|---|---|
| **Physical health** | | | | | | |
| Symptoms and functioning | 17 | 7 | 2 | 8 | 5 | 2 |
| Pain | 6 | 0 | 3 | 1 | 0 | 2 |
| **Mental health** | | | | | | |
| Broad mental health and wellbeing | 45 | 10 | 3 | 9 | 15 | 22 |
| Depression | 21 | 6 | 5 | 10 | 4 | 2 |
| Anxiety | 18 | 0 | 3 | 3 | 3 | 9 |
| Distress | 6 | 0 | 1 | 2 | 2 | 1 |
| **Quality of life** | 13 | 3 | 2 | 5 | 5 | 1 |
| **Social wellbeing** | | | | | | |
| Broad social wellbeing | 3 | 2 | 0 | 1 | 2 | 0 |
| Feelings of belonging | 16 | 0 | 0 | 2 | 3 | 11 |
| Connections and friendship | 18 | 0 | 0 | 3 | 5 | 10 |
| Loneliness and isolation | 21 | 0 | 1 | 5 | 3 | 12 |
| **Behaviour and decision-making** | | | | | | |
| Behaviour change | 16 | 1 | 2 | 1 | 4 | 9 |
| Motivation | 6 | 0 | 0 | 2 | 1 | 3 |
| Treatment adherence | 5 | 1 | 1 | 1 | 3 | 2 |
| Treatment decision-making | 18 | 1 | 0 | 1 | 11 | 7 |
| Self-efficacy | 22 | 3 | 3 | 7 | 5 | 8 |
| Empowerment | 10 | 0 | 1 | 1 | 2 | 6 |
| **Adjustment** | | | | | | |
| Illness acceptance | 16 | 5 | 0 | 2 | 7 | 7 |
| Feeling less alone | 17 | 0 | 0 | 1 | 4 | 12 |
| Feeling understood and reassured | 13 | 1 | 0 | 0 | 2 | 11 |
| Optimism and hope | 19 | 2 | 1 | 1 | 6 | 11 |
| Self-esteem | 12 | 0 | 0 | 0 | 7 | 5 |
| Post-traumatic growth | 2 | 0 | 1 | 1 | 0 | 0 |
| Identity | 7 | 0 | 0 | 0 | 1 | 6 |
| Coping | 18 | 4 | 1 | 6 | 6 | 5 |

pessimistic social comparison (e.g., fearing that future will be similar or feeling frustrated at own situation) negatively affected emotional wellbeing, whereas optimistic strategies (e.g., realising it is possible to improve or realising how well you are doing), did not negatively affect emotional wellbeing[86].

Six cross-sectional qualitative studies suggested that online support groups improved mental health as they provided a space to share experiences, receive and offer social support, have an outlet for feelings, and have opportunities to help others and learn new skills[26,45–47,66,73]. This was reported to be particularly important for those whose symptoms left them unable to carry out their usual purposeful activities[46]. Four also suggested that mental health improved due to improvements in social wellbeing and companionship[45,47,58,66]. On the other hand, seven suggested that poorer wellbeing, and increased fear, was influenced by exposure to negative aspects of conditions (including hospitalisation, relapses, suicidal thoughts, and death of other members), as well as complaints by other members and posts that were not solution-focused[22,46,73,79–82]. For some, positive stories were also damaging[69]. Qualitative studies also found that online support groups can focus too much on the condition[77], can be overwhelming in terms of the information[78,80], and can serve as a reminder for one's own negative health[79]. One also found that some participants experienced personal attacks or ridicule for their views and opinions, which led to feelings of mistrust and fear[78]. Feelings of frustration and disappointment were also reported in three (cross-sectional and intervention) qualitative studies, if participants were unable find online groups suited to their unique needs[82] and when having to wait for reply[72,87]. An interview study with Long Covid patients also found that some participants reported feeling frustrated or resentful of group members' who developed the condition after taking high-risk activities, such as travelling when it was advised not to[76].

### Depression

Outcome. Of five RCTs, one reported a reduction in depression over time in both an unmoderated and moderated Facebook group, with effects being sustained after one month[55]. However, three RCTs reported no differences in depression scores between online support groups (weekly moderated groups or a peer-led Facebook group – sometimes plus education), and control groups (education or usual care)[38,56,59]. Another longitudinal RCT reported no effect of time on depression in both a professionally moderated group and unmoderated group[88]. Furthermore, a pre-post intervention reported a reduction in depression following a 16-week intervention of weekly meetings combined with a private asynchronous newsgroup[57] but another pre-post survey found no difference in depression following a

## Table 5 | The role of usage characteristics on health outcomes

| Health outcome | Level of engagement (e.g., active vs passive user) | Intensity of usage (e.g., daily vs non-daily user) | Membership length |
|---|---|---|---|
| **Physical health** | | | |
| Symptoms and functioning | Higher engagement and better symptoms and functioning [39,49] Higher engagement and reduced symptoms and functioning [28,40] No relationship between engagement level and symptoms and functioning [28,39,40,49,53]. | Less intensity of use and better symptoms and functioning [148] No relationship between intensity of use and symptoms and functioning [49,148] | |
| Pain | No relationship between engagement and pain [28,55]. | No relationship between intensity of use and pain [148]. | |
| **Mental Health** | | | |
| Broad mental health and wellbeing | No relationship between engagement level and broad mental health outcomes [28,49,61,62,86]. | No relationship between intensity of use and broad mental health outcomes: [49,61,62,148]. | |
| Depression | Higher engagement and increased depression scores [53] No relationship between engagement and depression [28,40,55,61,62,86] | Higher intensity and better depression scores [123] No relationship between intensity of use and depression [61,62,91] | Longer duration and better depression scores [123,146] No relationship between duration and depression [91] |
| Anxiety | No relationship between engagement and anxiety [55] | No relationship between intensity of use and anxiety [91] | Longer duration and better anxiety scores [146]. No relationship between duration of use and anxiety [91] |
| Distress | No relationship between engagement and distress [28] | | |
| **Quality of life** | | | |
| Quality of life | No relationship between engagement and quality of life [28] | | Longer duration of use and better quality of life [146] |
| **Social Wellbeing** | | | |
| Social wellbeing | Higher engagement and better social wellbeing [28,151] No relationship between engagement and social wellbeing [53] | Higher intensity of use and poorer social wellbeing [148] | Longer duration of use and better social wellbeing [102] |
| Loneliness and isolation | No relationship between engagement and loneliness [28] | Higher intensity of use and reduced loneliness [123,147] | |
| Connections and friendships | Higher engagement and better connection outcomes [49] No relationship between engagement and connection [40] | Higher intensity of use and better outcome [49] | |
| **Behaviour** | | | |
| Behaviour change | Higher engagement and better behavioural outcomes [49] No relationship between engagement and self-management [150] | Higher intensity of use and better behavioural outcomes [49] | |
| Treatment decision-making | Higher engagement and better treatment-decision making outcomes [49,119], | Higher intensity of use and better treatment decision-making outcomes [49,91,119]. No relationship between intensity and treatment decision-making [91]. | Longer membership duration and better treatment confidence [102] No relationship between membership duration and treatment decision-making [91]. |
| Self-efficacy | Higher engagement and higher self-efficacy [39,101] No relationship between engagement and self-efficacy [28] | Higher intensity of use and higher self-efficacy [101] | Longer duration of use and better self-efficacy outcomes [146] |
| Empowerment | Higher engagement and better empowerment outcomes [49] | Higher intensity of use and better empowerment outcomes [49]. | |
| **Adjustment** | | | |
| Illness acceptance | No relationship between engagement and illness acceptance [28,151] | | No relationship between membership duration and illness acceptance [102] |
| Feeling less alone | Higher engagement and feeling less alone [49] | Higher intensity of use and feeling less alone [49] | |
| Feeling understood | Higher engagement and feeling more understood [49]. | Higher intensity of use and feeling more understood [49]. | |
| Self-esteem | No relationship between engagement and self-esteem [151] | | No relationship between duration of use and self-esteem [102] |
| Optimism and hope | No relationship between engagement and optimism and hope [28,151] | No relationship between intensity of use and optimism and hope [91,127] | Longer duration of use and better optimism scores [123] No relationship between duration of use and optimism and hope [91,102,127] |
| Coping | No relationship between engagement and coping [28] Higher engagement and better coping [28]. | Higher intensity of use and better coping outcomes [148]. No relationship between intensity of use and coping [91,148] | Longer membership duration and better coping outcomes [91] |

6-month WhatsApp group intervention [89]. Furthermore, a non-randomised controlled trial found improvements in depression scores following weekly moderated sessions, but no differences post-intervention between the intervention and treatment as usual [60]. Similarly, two non-randomised control trials compared depression between participants in moderated weekly video groups and control groups (journalling or no treatment) and found no differences between conditions [41,90], or over time [90]. Additionally, a longitudinal intervention reported no differences in depression scores

**Table 6 | The relationship between group type and health outcome**

| Group type | Finding |
|---|---|
| Professionally moderated vs unmoderated groups | In a longitudinal RCT, pain, depression, anxiety, and self-efficacy scores increased in both a researcher-moderated and unmoderated Facebook group [55,121]. Similarly, a longitudinal RCT found no effect of group type (professional vs peer-led) on depression scores[88]. In a cross-sectional survey, more than 85% of participants reported being happy that the online support group was peer-to-peer (i.e., run by regular people 'like me' who have also experienced the condition) rather than trained professionals (e.g., therapists, doctors or nurses)[70]. Similarly, another cross-sectional survey found that self-efficacy was associated with medical adherence in both healthcare professional groups (i.e., established by hospitals or healthcare professionals) and non-healthcare professional groups, but this association was stronger in non-healthcare professional groups[152]. |
| Online vs face-to-face groups | In one cross-sectional survey, there was no difference in depression or anxiety outcomes between those attending an online or face-to-face group, but those attending a face-to-face group reported more positive wellbeing and less distress[43]. Two cross-sectional surveys reported no differences in treatment decision-making between participants in an online or face-to-face group [43,92]. Furthermore, participants in another cross-sectional survey reported that they felt more comfortable in an online group, compared to face-to-face, as they knew no-one was looking at them when they shared their stories / feeling / problems or asking questions[70]. An interview study found that whilst some participants wanted to attend in person groups, they are often unable to attend due to the severity of their symptoms therefore find online groups can be advantageous[46] |
| Local groups (groups with a focus on a local area, rather than groups with an international or non-specific geographical focus) | Local groups provided opportunities for patients to connect and maintain a sense of belonging offline, as well as in the online groups[47] and some joined such groups with the intention of meeting in person one day[46] |

between users and non-users at six weeks or 3 months[40] and a longitudinal survey also found no change in depression over time[61]. In one cross-sectional survey, 55% of participants reported improvements in depressed feelings[91], but two cross-sectional surveys found no difference in depression scores between an online and face-to-face group[43,92]. However, a cross-sectional interview found that some participants feel more depressed after reading negative posts on a Facebook group[93].

Mechanisms. In terms of content expressed, three analyses of posts made on online support groups within experimental and naturalistic settings reported no association between each of empathy expression[94], religious expression[53], or insightful disclosure[50] with depression. With regards to support and comparison, one cross-sectional survey reported that depression was negatively predicted by social support[95], but another found that receiving and offering emotional support and receiving advice was not associated with depression[96]. With regards to social comparison, one cross-sectional survey found that upward contrast negatively predicted depression (but not downward identification, upward identification or downward contrast)[86]. In a cross-sectional survey, depression scores amongst passive users was not associated with conflict (e.g., feeling burdened or misunderstood) or universality (e.g., findings others with similar experiences), but conflict was positively associated with depression for active users[96].

### Anxiety
Outcome. Of three RCTs, one reported reductions in anxiety scores in both a moderated and unmoderated Facebook group, but these effects were only sustained 1-month post-intervention in the unmoderated group, not the moderated group[55]. However, two RCTs found no difference in anxiety scores over time between the online support groups (moderated synchronous text-based sessions or moderated Facebook group) plus education and an educational control[38,59]. Furthermore, qualitative and quantitative findings of quasi-experiments reported a reduction in anxiety in a moderated synchronous online support group compared to a no-treatment control group[41] and following an unmoderated email-based support group[97]. However, another quasi-experiment reported whilst anxiety scores improved over time, there were no differences between a moderated synchronous weekly chat group and treatment as usual[60]. Two cross-sectional surveys found no difference in anxiety scores between participants in an online or face-to-face support group[43,92], with nearly 60% of participants in one study reporting improved anxious feelings [126]. Four cross-sectional qualitative studies reported a reduction in anxiety[47,78,91,98]. However, five

qualitative studies reported the potential for online support groups to increase anxiety[72,76,77,82,93], with this causing some individuals to limit their usage of the group[77].

Mechanisms. A quantitative cross-sectional survey found that giving and receiving emotional support and receiving advice were negatively correlated with anxiety for active users, whereas receiving advice, and universality (e.g., finding others similar to you) was negatively correlated with anxiety for passive users[96]. Qualitative findings suggested that online support groups quelled anxiety as they helped to manage unfamiliar symptoms and provided emotional and informational support[47,78,97,98]. However, they also suggested that online support groups may increase anxiety after reading 'horror stories' and messages that can bring attention to specific issues that could be faced in the future[72,77,82,93].

**Distress**. Distress refers to distress from traumatic events and emotional distress more generally.

Outcome. An RCT compared an unmoderated email group with an educational website and found no difference in distress scores over time or between groups[37]. However, whilst one quasi-experiment found that distress significantly decreased over time amongst participants in a moderated synchronous online support group, there were no differences between the intervention and treatment as usual control[60]. Similarly, a non-randomised controlled trial found no differences in distress between a moderated synchronous group and a no-treatment control[41]. Furthermore, a cross-sectional survey found that 100% of participants reported that private email groups helped them deal with their emotional distress[70]. Another cross-sectional survey reported that distress was less frequent in a face-to-face group than an online support group[43]. Qualitative findings also suggested that seeing others' stories can lead to increased distress[26].

Mechanisms. Participants in a cross-sectional qualitative survey reported that distress increased when the posts are skewed to sad or negative[26]. This study also found that positive stories can be distressing, for example reading members' pregnancy stories can be difficult for those with fertility issues[26].

### Quality of life
The following studies refer to a broadly measured quality of life; where subscales of quality of life are reported (e.g., role functioning) these are reported in their respective section.

https://doi.org/10.1038/s44271-025-00217-6                                                                    **Article**

## Table 7 | Summary of findings

| Category | Health and Wellbeing Outcome | Summary of findings |
|---|---|---|
| **Physical health** | Symptoms and functioning | **Summary**: there is mixed evidence for the impact of online support groups on symptoms and functioning. While some studies report improvements, others found no change.<br>**Mechanisms**: informational support and sharing experiences may positively influence symptoms and functioning. |
| | Pain | **Summary**: the evidence from cross-sectional and experimental studies suggests that online support groups may be able to help with group members' pain. |
| **Mental health** | Broad mental health and wellbeing | **Summary:** the effects of online support groups on mental health is mixed. While many studies report improvements in mental health, more report either no or negative effects.<br>**Mechanisms:** influencing factors for positive wellbeing include receiving emotional support, having an outlet for feelings, sharing experiences, helping others and through improved social wellbeing. Influencing factors for negative wellbeing include negative social comparison, information overload, reading emotional information, unsuitable groups and lack of replies. |
| | Depression | **Summary:** no study reported a negative effect of online support groups on depression, but it is unclear whether online support groups have a positive or neutral effect on depression.<br>**Mechanisms:** receiving emotional support may positively influence depression scores, whilst negative social comparison can negatively influence it. Some studies also highlight the complexity of factors such as emotional expression. |
| | Anxiety | **Summary:** there was mixed evidence regarding the impact of online support groups on anxiety with studies reporting, positive, negative and no effects.<br>**Mechanisms:** anxiety can be mitigated through emotional and informational support which helps manage unfamiliar symptoms, but it can be heightened through reading horror stories. |
| | Distress | **Summary:** findings are mixed with studies reporting positive, negative and no effects on distress.<br>**Mechanisms:** distress may increase when posts are skewed to be sad or negative. |
| **Quality of life** | Quality of life | **Summary:** no studies reported a negative effect of online support groups but the evidence is mixed relating to a positive or neutral effect on quality of life.<br>**Mechanisms:** emotional support may enhance quality of life. |
| **Social wellbeing** | Broad social wellbeing | **Summary:** one study looked at social wellbeing more broadly and found that 52% of participants reported improvements.<br>**Mechanisms:** factors associated with enhanced social wellbeing include emotional support. |
| | Feelings of belonging | **Summary:** evidence, from mostly qualitative data, suggests that group members feel a sense of belonging to the online support group. However, when it is hard to join in conversations or there is no response to message one study found that it can lead to people feeling like an outsider and sometimes leaving the group.<br>**Mechanisms:** being part of a group with people living with the same condition, which helps people feel part of a group, have discussions and develop a shared identity. |
| | Connections and friendship | **Summary:** many members of online support groups form new friendships and social connections, but this is not the case for everyone and can be difficult to do.<br>**Mechanisms:** positive association between perceived credibility and competence of discussion on online communities and social capital within online groups. |
| | Loneliness and isolation | **Summary:** the evidence on isolation and loneliness outcomes is mostly positive but there is also the potential for feelings of isolation to resurface when logging off from the groups.<br>**Mechanisms:** being an active member and making new friends positively influences loneliness and isolation. |
| **Behaviour and decision-making** | Behaviour change | **Summary:** the evidence suggests that online support groups can encourage positive behaviour change such as engaging in preventative behaviours, changing risky behaviours, purchasing assistive devices and trying other people's dietary habits.<br>**Mechanisms:** behaviour change may be facilitated by reading others' experiences and through the shared advice and the perceived credibility of such discussions. |
| | Motivation | **Summary:** studies suggest that participation in online support groups may increase motivation to make a positive lifestyle change or keep up with self-management.<br>**Mechanisms:** motivation may be influenced by sharing experiences, seeing success stories, and receiving non-judgmental advice. |
| | Treatment adherence | **Summary:** most studies suggest no changes in adherence after participation in an online support group<br>**Mechanisms:** treatment adherence may be influenced by sharing experiences and discussing treatments and medication with others. |
| | Treatment decision-making | **Summary:** findings suggests that online support groups may influence treatment decision making (e.g., changing initial treatment and assessing benefits and side-effects), although there is large variability in the proportion of participants reporting this.<br>**Mechanisms:** sharing experiences and information and receiving emotional support may influence treatment-decision making. Intensity of usage may also influence treatment decision making (i.e., being a daily user) but this depends on the measurement. |
| | Self-efficacy | **Summary**: most studies reported positive effects of online support groups on self-efficacy, although there is large variation in the proportion of participants reporting improvements and some studies reported no changes over time, nor any added benefits compared to educational controls.<br>**Mechanisms:** receiving informational and emotional support, helping others, religious expression and using positive emotion words has been found to be associated with self-efficacy. |
| | Empowerment | **Summary:** most studies report improvements in empowerment.<br>**Mechanisms:** feeling empowered may be related to the information shared and being part of a collective voice as it enables group members to feel in control. |

**Table 7 (continued) | Summary of findings**

| Category | Health and Wellbeing Outcome | Summary of findings |
|---|---|---|
| Adjustment | Illness acceptance | **Summary:** findings suggest that online support groups may help group members accept their illness and have a positive appraisal of their condition.<br>**Mechanisms:** illness acceptance may be influenced by social support, comparison with others and findings others in a similar situation. |
| | Feeling less alone | **Summary:** the evidence suggests that group members may feel less alone by being part of an online support group.<br>**Mechanisms:** participants reported feeling less alone after seeing others having similar feelings, emotions and experiences and receiving emotional support. |
| | Feeling understood and reassured | **Summary:** the evidence suggests that online support groups may help group members feel understood and reassured<br>**Mechanisms:** participants felt understood because of the shared experience and such feelings were associated with perceived credibility of discussions. |
| | Optimism and hope | **Summary:** the evidence, from mostly qualitative studies, suggests that members of an online support group feel optimistic and hopeful towards the future after using a group.<br>**Mechanisms:** optimism and hope are influenced by reading success stories, positive comparison, receiving emotional support and finding positive meaning. |
| | Self-esteem | **Summary:** most studies suggest that online support groups may enhance self-esteem<br>**Mechanisms:** studies suggest that self-esteem is associated with emotional and social support. |
| | Post-traumatic growth | **Summary:** the evidence suggests no change in post-traumatic growth. |
| | Identity | **Summary:** the evidence suggests that participating in online support groups may help group members' personal and group identities as they rediscover their sense of self, return to a lost version of themselves, feel normal again and connect with others.<br>**Mechanisms:** sharing humour, being part of a majority and illness acceptance may facilitate changes in identity. |
| | Coping | **Summary:** the findings are mixed regarding the effect of online support groups on coping. While many studies report a positive influence, some report either a negative or no effect on coping.<br>**Mechanisms:** Connecting with others who understanding, being accepted and receiving social support may positively influence coping. |

**Outcomes**. One RCT and two quasi-experiments found no differences between an online support group (private Facebook group or moderated synchronous groups) and control (education, usual care, or no treatment)[38,41,56]. Another RCT found that whilst quality of life scores improved over time following a moderated synchronous group, there were no differences between the intervention and treatment as usual[60]. However, 100% of (seven) participants agreed that the posts in a secret Facebook page were helpful in improving their quality of life[42]. Similarly, a cross-sectional quantitative survey reported that 94.7% in a private email group said that the group made a positive difference to their quality of life[70] and another cross-sectional survey reported lower quality of life scores in a face-to-face group than an online support group[43].

**Mechanisms**. Giving and receiving informational support was not associated with quality of life in a cross-sectional survey and a content analysis within an intervention study[99,100], whereas perceived emotional support was, with this outcome being mediated by contentment[99]. However, in cross-sectional surveys, existential quality of life was associated with receiving online social and emotional support[48], companionship[47], and relatedness[47], but not online informational support[47]. A content analysis within a longitudinal survey of new members of an existing asynchronous bulletin board found that insightful disclosure was not associated with quality of life scores[50]. Furthermore, a cross-sectional qualitative survey reported that their quality of life had improved through the support from group members[101]. A cross-sectional quantitative study found no association between perceived competence of discussions within an online support group and quality of life[49].

*Social wellbeing*
Social wellbeing outcomes include broad social wellbeing, feelings of belonging, connections and friendship, and loneliness and isolation.

Broad social wellbeing
Outcome. One cross-sectional study found that 52% of participants reported enhanced social wellbeing from being part of an online support group[102].

Mechanisms. Two cross-sectional surveys suggested found that exchanging social support[23,102] and encountering emotional support[23] were positively associated with social wellbeing. However, there were conflicting findings for sharing experiences as whilst one cross-sectional study found that it was positively associated with social wellbeing[23], another reported that it was not[102]. A cross-sectional survey and a quasi-experiment reported that enhanced social wellbeing was not predicted by use of religious expression[53], information exchange[102], helping others[102] or comparison with others[102].

Feelings of belonging
Outcome. A pre-post intervention survey had mixed results as although women in an unmoderated email group agreed that they felt a sense of belonging, some also reported leaving groups as they felt different from other members and did not feel close to the group[97]. Furthermore, in interviews following a non-randomised controlled trial, participants reported finding community in the weekly professionally moderated support group[41]. Similarly, 90% of participants in a cross-sectional survey reported a sense of belonging as a result of comments or posts from other members[69] and similar findings were reported in two more cross-sectional surveys[66,68] and all 11 cross-sectional qualitative studies[22,45–47,49,74,75,77,80,82,103]. However, two cross-sectional qualitative studies also reported that some group members do not feel a sense of belonging within the group[46,77].

Mechanisms. Qualitative findings suggested that feelings of belonging arose from interactions with others and were attributed to the common ground amongst group members and to being part of a group of people living with the same condition, which helped group members to fit in, have discussions and develop a shared identity[45,46,75,80,82]. However, another interview study found that some participants felt like outsiders due to difficulties in joining conversations, receiving no, or unhelpful, responses, finding posts too negative or too positive, or feeling like their needs are not represented within the group[46,77].

Connections and friendship
Outcome. The quantitative findings of an intervention study did not find an increase in the number of friendships following a combined synchronous and asynchronous online support[104]. However, in post-intervention

interviews, participants reported experiencing improved relationships and being more confident in their ability to make and socialise with friends following the combined synchronous and asynchronous group[104] and amongst participants of an unmoderated email group[97]. Furthermore, a longitudinal intervention reported no differences in bonding scores between users and non-users at six weeks or 3 months[40]. Four cross-sectional surveys reported that between 44 and 66% of participants formed new friendships in asynchronous groups[23,49,105,106] and another found that 94.7% bonded with the other women in an email group[70]. This is echoed in all 10 cross-sectional qualitative studies, as participants reported developing true friendships and bonds and felt connected to others[22,46,47,76,78,80,81,101,107,108]. With regards to offline relationships, sometimes new social contacts replaced friendships lost because of their condition[22], sometimes they supplemented existing offline friendships[22], and other times they led to a decline in real-life relationships due to being over-reliant on online relationships and decreased attention to offline relationships[78]. Furthermore, participants in two interview studies reported difficulties forming new relationships[78,108].

Mechanisms. Participants connected with others through similar diagnoses, symptoms, illness management issues, as well as personal characteristics such as sense of humour[46,108]. Participants felt connected to other members through the conveyed emotion, although some participants found this difficult due to the lack of body language and not being an active member[80,108]. Furthermore, some participants found it difficult to connect to those with different experiences, such as those who are newly diagnosed, have more severe disabilities, have less family support, or do not share the same political interests[108]. Furthermore, one cross-sectional qualitative study found that although group members felt sad when a fellow member passes away they also felt more connected to each other[81]. Another found a positive correlation between the perceived credibility and competence of discussion on online communities and social capital within online groups[49].

**Loneliness and isolation**. Loneliness and isolation refers to the feelings following the formation of friendship and connections and is distinguished from feeling less alone (included in the adjustment sections) following seeing others with similar experiences. Loneliness and isolation outcomes have been grouped together, despite the differences in definitions[109], as the terms are used interchangeably within the included studies to refer to an absence or presence of social connections.

Outcome. An RCT compared a moderated synchronous group plus educational website to the website alone and found better loneliness scores in the online support group condition[59]. The quasi-experimental studies reported conflicting results; quantitative findings from a post-intervention study found reductions in loneliness scores after a 12-week synchronous chat session intervention[104], which is echoed in post-intervention interviews of the same study as well as after a four-month unmoderated email group[97]. However, three quasi-experiments reported no effects online support groups (combined synchronous and synchronous groups or moderated synchronous group alone) on loneliness over time[90,103] or compared to either a no treatment, or active, control[41,90]. On the other hand nine cross-sectional qualitative studies[22,26,45,46,74,76,78,82,108] and three cross-sectional quantitate surveys reported reductions in isolation, with this being reported in 47-75% of participants[23,69,106]. However, two cross-sectional qualitative studies suggested that participants sometimes felt isolation within an online support group[80] and after logging off[72].

Mechanisms. Qualitative studies suggest that reductions in isolation occurred by connecting with others, making new friends, feeling part of a group and becoming more outgoing[22,26,45,46,74,76,82,104]. This was often particularly needed as the physical constraints of chronic conditions make it difficult socialise[108]. However, participants can feel isolated in online support groups as they lack human touch and connection[80] or because they feel different from others, which can result in them leaving groups[97].

## Behaviour and decision-making
This includes behaviour change, motivation, treatment adherence, treatment decision-making, self-efficacy and empowerment.

### Behaviour change
Outcomes. The quantitative findings of an RCT, comparing an online support group plus an educational website to an educational website alone, found no difference in behaviours relating to disease management or health promotion between the groups after the intervention[38]. However, the qualitative findings of another RCT suggested that participants tried new things and were more active after using the online support group[87]. Similarly a pre-post survey following a WhatsApp group intervention found improvements in behaviour[89]. Similarly, post-intervention interviews following a quasi-experiment suggested that participants learned tips to help with their day-to-day life (e.g., where to place an inhaler)[103]. Furthermore, three cross-sectional surveys reported mixed findings. One found higher scores for self-management of diabetes amongst participants not belonging to an online support group compared to online support group members[110], whereas two reported improvements for those who had participated in groups. Specifically, one reported increased odds for lifestyle changes for those who had participated in online support groups in the previous year[111], whilst another reported improvements in self-management and adopting a healthy lifestyle for those in a virtual online community[68]. Furthermore, nine cross-sectional qualitative studies suggested that upon joining an online support group, participants gained the skills for self-management of their condition and started taking better care of themselves (e.g., engaged in preventative activities, changed risky behaviours, purchased assistive devices, and tried other people's dietary habits)[22,45,58,69,72,78,101,112,113].

Mechanisms. Five qualitative studies suggested that behaviour change was possible after reading about the experiences of others and through sharing advice in online support groups[45,58,72,112,113], and a quantitative survey found that credibility of discussion on online communities positively correlated with self-care[49].

### Motivation
Outcome. Interviews following a quasi-experiment, including a moderated discussion forum plus education (compared to education alone), suggested that participants were motivated to keep up with self-management[87], but a pre-post survey found no differences in motivation to adhere to HIV treatment following a WhatsApp group intervention[89]. A cross-sectional survey reported mixed findings on motivation outcomes, as it found an increase in motivation scores amongst participants with Type 2 diabetes but a decrease amongst those with Type 1[63]. Moreover, three cross-sectional qualitative studies reported an increase in motivation to change behaviour[45,75,112].

Mechanisms. Post-intervention interviews suggested that participants were motivated to keep up with self-management after reading posts of other people who were still active despite their pain[87]. This was echoed in cross-sectional qualitative studies which reported that motivation was influenced by seeing other people make healthy lifestyle choices, sharing success stories and receiving non-judgmental personalised advice[45,75,78,112].

### Treatment adherence
Outcome. One RCT reported no effects of support group membership on medication and infection control adherence, within and between conditions[114]. This is supported by post-intervention interviews following a WhatsApp group intervention[89] and a cross-sectional survey which found that social networking support group membership was not related to self-reported infection control adherence[115]. However, improvements in medication adherence were reported by online support group users in an interview and Delphi study, in the same paper[45].

Mechanisms. Two qualitative studies suggested that treatment and medication adherence was facilitated by observing similar patients' health status, sharing (positive and negative) experiences and being able to discuss with others (e.g., tracking and side-effects)[45,116]. However, a cross-sectional survey found no relationship between perceived social support from online peers and reported medical adherence[115].

**Treatment decision-making.** Treatment decision-making refers to group members' ability to make decisions relating to their treatment, revising their initial treatment plan and feeling confident in their treatment.

Outcome. In interviews following an unmoderated email-based support group intervention, participants reported being more active in terms of their treatment[97]. Four cross-sectional surveys report that group members learn about existing (50-60%) and alternative (60%) treatments; received treatment advice (20%); and can feel more confident in their chosen treatment[68,92,102,117,118]. Six cross-sectional quantitative studies (including one Delphi study) reported that 25-80.5% of participants reported learning about new treatments, having their treatment requests influenced by an online support group, or choosing to change their initial treatment after participating in an online group[43,45,91,92,119,120]. When comparing to face-to-face support groups, two cross-sectional surveys found no differences in treatment decision-making outcomes between the groups[43,92]. Furthermore, seven cross-sectional qualitative studies reported feeling empowered in relation to treatment decision-making and feeling more confident in their treatment[22,26,45,58,74,77,112].

Mechanisms. Four qualitative studies reported that support with treatment decision-making occurred through connecting with other group members and sharing experiences and information as it allowed members to assess the benefits and side-effects of treatment and identify best practice[22,26,45,97]. With regards to treatment confidence, two cross-sectional quantitative surveys reported that social comparison[102] and finding recognition[23] predicted treatment confidence. However, there was conflicting evidence regarding the role of receiving emotional support, as although one cross-sectional survey found that it predicted treatment confidence[23], another did not[102]. These two surveys also reported that treatment confidence was not predicted by information exchange, helping others or sharing experiences[23,102].

**Self-efficacy**
Outcome. Two RCTs reported improvements over time in moderated and unmoderated Facebook groups (sometimes plus education)[38,121], although one reported no differences between participants in the online support group and educational control[38]. However, another RCT found that emotional self-efficacy declined amongst participants in an unmoderated email group[37]. A non-randomised controlled trial reported no differences in self-efficacy between a weekly professionally moderated support group and a no-treatment control[41]. Furthermore, a post-intervention survey found no difference over time following a 12-week synchronous online support group[104], but a pre-post survey found improvements in adherence self-efficacy following a WhatsApp intervention[89]. Other post-intervention interviews reported improvements after an 8-week synchronous online support group supplemented with gamification communication[103]. A longitudinal survey, three cross-sectional surveys and a Delphi study reported improvements in self-efficacy amongst participants, but there was variation in the proportions of people reporting such an effect (19.1–88.5%)[39,45,68,71,102]. Eight cross-sectional qualitative studies also reported improvements in self-efficacy[22,26,45,46,49,77,101,112].

Mechanisms. With regards to content expressed on online support groups, two content analyses within intervention studies and a cross-sectional survey found that writing a higher number of religious expressions[51], using more positive emotion words[52], receiving social support[122] and helping others[122] was associated with improved self-efficacy, but disclosing negative emotions was not[52]. Similarly, qualitative studies found that the information and support on online support groups enabled people to take an active role in managing their condition and feel like they can regain control over their personal lives[22,46,77,112].

**Empowerment**
Outcome. An RCT comparing a peer-led Facebook group plus online education to education alone found no differences in empowerment at 3 or 6 months[38]. However, post-intervention interviews following an unmoderated email-based support group found that participants felt empowered following the intervention[97]. Furthermore, across two quantitative studies between 73-80.7% of participants reported that online support groups improved empowerment[45,49]. Six cross-sectional qualitative studies also suggested that participants feel more empowered by being part of an online support group[26,49,75,98,107,112].

Mechanisms. Qualitative studies suggested that feeling empowered was mostly in relation to the information shared, which enabled group members to feel in control[26,75,97,107]. Participants also reported feeling empowered by helping others[49,112] and being part of a collective voice[98]. A quantitative study reported that empowerment was positively associated with perceived credibility of discussions on online communities and behaviours such as requesting or sharing informational and emotional support[49].

**Adjustment**
This section includes illness acceptance, feeling less alone, feeling understood and reassured, self-esteem, optimism and hope, post-traumatic growth, identity, and coping.

**Illness acceptance**
Outcome. Two cross-sectional surveys reported that approximately 30% of participants said that the online support group helped them find meaning in their experience[71] and improved acceptance of their condition[102]. However, another cross-sectional survey reported that face-to-face support group members accepted their illness better than those in online support groups[43]. Seven qualitative studies also found that online support groups helped group members to accept their illness[22,26,72,113], view it more positively[72], reappraise it as something that can be successfully managed[72], overcome its uncertainty[98], conceptualise the illness as chronic rather than terminal[97], and allowed members to understand their condition as defined by the community[113].

Mechanisms. Four qualitative studies suggested that illness acceptance was facilitated by emotional expression[74], comparison with other group members (particularly those with more severe symptoms)[72,113] and finding others in a similar situation[26]. Two cross-sectional surveys and two content analyses within interventions found that illness acceptance and positive reframing were not associated with empathy reception[85], receiving emotional/social support[23,85,102], information exchange[23,102], helping others[23,102], finding recognition[23], sharing experiences[23,102] or religious expression[51]. On the other hand, across three cross-sectional quantitative studies positive reframing and illness acceptance was positively associated exchanging social support[85,102], empathy expression[85] and comparison with others[102,123]. Additionally, a cross-sectional survey found that those who were inhibited from making contributions to online support groups because they either felt a poor sense of community or had concerns about privacy and disclosure were less likely to feel they had found positive meaning from the online support groups[124].

**Feeling less alone**
Outcome. Post-intervention interviews in one quasi-experiment reported that participants felt less alone following the intervention[103]. Four cross-sectional quantitative[43,49,70,105] and 12 cross-sectional qualitative studies[22,46,58,72,74,76,78,82,108,112,125,126] also reported that participants felt less alone. In the surveys, this occurred in 76-100% of participants.

**Mechanisms.** Participants in cross-sectional qualitative studies reported feeling less alone as they can connect with others[74], receive support[105], compare to other group members knowing that others have similar feelings, emotions and experiences[46,76,78,112,126], and have shared understanding and empathy amongst group members[82]. Online support groups are particularly beneficial for connecting those with rare conditions, and helping them to feel less alone[108]. A cross-sectional survey also found that feeling less alone was also positively associated with perceived credibility of discussions on online communities and behaviours such as requesting or sharing informational and emotional support[49].

### Feeling understood and reassured
**Outcome.** One cross-sectional survey reported that 15% of participants reported that they felt reassured in a moderated asynchronous online support group[106]. All (11) cross-sectional qualitative studies reported that online support groups enabled participants to feel understood and reassured[22,26,49,69,72,73,76,78,79,87,108]. Three of the qualitative studies reported that online support groups reassured group members that they were not 'crazy' and that their symptoms were not 'psychosomatic'[22,73,76].

**Mechanisms.** Qualitative studies suggested that participants felt understood and reassured because of the shared experience[22,108], peer support[79] and reading others' experiences, particularly those who share similar symptoms[26,72,73,78,79,87]. A cross-sectional survey found that feeling understood was positively associated with perceived credibility of discussions on online communities and behaviours such as requesting or sharing informational and emotional support[49].

### Optimism and hope
**Outcome.** One RCT found a deterioration in hope after 4 months in an unmoderated email group, but no differences were found between the online support group and an educational website[37]. Conversely, post-intervention interviews following a quasi-experiment suggested that an unmoderated email-based support group increased hope[97]. Furthermore, four cross-sectional surveys reported increases in optimism and hope, reporting that between 19% and 75% of participants experienced improvements[69,71,91,102]. All (11) cross-sectional qualitative studies reported increases in optimism and hope[22,45,47,69,72,73,75,77–79,107] but two also suggested decreases in these outcomes[69,77].

**Mechanisms.** 10 qualitative studies highlighted the importance of reading success stories and comparing to other group members, particularly those who have had the condition for longer and are improving[45,47,72,73,75,77–79,97,107], with one qualitative study suggesting that other members serve as positive role models[22]. Cross-sectional surveys suggest that receiving emotional and social support, finding recognition, comparison with others, and positive meaning may predict optimism and hope[23,102,123,127]. However, one cross-sectional survey reported that exchanging support and sharing experiences did not predict optimism and hope[102]. There was conflicting evidence between three cross-sectional surveys regarding receiving information and helping others, as two found that these factors did predict optimism[102,123], but another found that they did not[23].

### Self-esteem
**Outcome.** Six cross-sectional quantitative studies reported that between 26% and 88.4% of participants experienced improvements in self-esteem and self-confidence[23,43,45,69,102,106]. However, another cross-sectional survey found no difference between those who use Facebook forums and those who do not[63]. Five qualitative cross-sectional studies also report enhanced self-esteem and self-confidence[22,45,72,107,128].

**Mechanisms.** Two cross-sectional qualitative studies suggested that enhanced self-esteem was facilitated by receiving appreciation from other group members, through the gratification they felt from being active online, and from giving back to the group by sharing personal experiences[22,107]. Two

cross-sectional quantitative surveys suggest that self-esteem was not associated with information exchange[23,102], finding recognition[23], comparison with other members[102], helping others[23,102] or sharing experiences[23,102], but may be predicted by encountering emotional support[23] and exchanging social support[102].

### Post-traumatic growth
**Outcome.** An RCT and quasi-experiment found no changes over time in post-traumatic growth amongst participants in a weekly synchronous online support group compared to usual care[56] and scores prior to the intervention[57].

### Identity
**Outcome.** One cross-sectional survey reported that 93.1% of participants said that group participation had helped them recover their sense of self[70]. Similarly, two cross-sectional qualitative studies reported that participants formed new identities through accepting the changes that come with their condition and by returning to a lost version of themselves[47,74]. Four qualitative studies also reported that participants felt "normal" again after participating in the online support group[58,82,112,116].

**Mechanisms.** Across six qualitative studies, three reported that they felt "normal" again as their experiences were normalised[82], they were part of a majority (vs being an outlier)[58], and they shared gallows humour[112]. Two also reported that participants formed new identities through accepting the changes that come with the condition[74] and feeling connected to a group[47].

### Coping
**Outcome.** One RCT found better coping outcomes in an educational control compared to a 12-weekly moderated online support group during the intervention[59]. However, after the intervention, coping outcomes on one sub-scale (self-blame) were more favourable in the online support group condition. A pre-post intervention study found an increase in support-seeking coping following weekly synchronous groups supplemented with a gamification social setting, with these quantitative findings echoed in the qualitative evaluation[103]. However, in a similar study by the same research team, they found no differences in coping scores following a 12-week moderated synchronous online support group, but post-intervention interviews suggested that participants sought more support-seeking coping strategies after the intervention[104]. Furthermore, another pre-post study reported reduced coping following a combined synchronous and asynchronous online support group[57] and a non-randomised controlled trial found improvements in coping following weekly moderated sessions, but not differences post-intervention between the intervention and treatment as usual[60]. Four cross-sectional quantitative surveys reported that between 60% and 88.1% of participants found that the online support group helped them to cope with their condition[69,71,91,92], although one study found that a higher proportion of participants reporting increased coping in face-to-face groups[92]. Five cross-sectional qualitative studies reported coping outcomes with all suggesting that online support groups help people cope with their condition[22,26,45,74,108]. One interview study reported that 82.7% of participants found that online interactions helped them learn how to cope with the social, physical and health consequences of the diseases[45]. However, another interview study reported that there are limitations in the extent to which the groups can help as participants recognise that the groups do not substitute the support from health professionals[108].

**Mechanisms.** Qualitative studies suggested that coping was facilitated by connecting with other people who understand[26,108], having individual differences accepted[74], and receiving social support[45]. Across two cross-sectional quantitative studies and two content analyses within intervention studies, giving and receiving informational support, empathy reception, social support, and finding positive meaning were positively associated with adaptive coping[94,95,100,122], whereas helping others and empathy expression were not[94,100].

## Discussion

This systematic review sought to investigate whether health and wellbeing outcomes are influenced by participating in online support groups for chronic conditions. We also sought to identify the factors influencing such outcomes. Summarising the findings of 100 papers, health outcomes were categorised as physical health, mental health, quality of life, social wellbeing, behaviour and decision-making, and adjustment, which broadly aligns with outcomes from a recent umbrella review exploring other types of peer support for people with chronic conditions[129]. The sections below, organised by research question, summarise, and discuss the findings.

### What are the effects of online support groups on the observed and self-reported health and wellbeing of individuals with a chronic condition?

The findings suggest that online support groups may positively influence pain, social wellbeing, adjustment and behaviour change and decision-making. By surrounding oneself with people with the same condition within an online support group participants reported feeling less alone and more understood, reassured and optimistic. Similarly, participants reported that online support groups helped them find meaning, feel "normal" and either re-discover their old sense of self or discover a new identity, which is particularly important as those with chronic conditions often experience a loss of personal identity[130]. In addition to changes in identity, people with chronic conditions reported experiencing a loss of social connections upon their diagnosis[131]. Cross-sectional quantitative and qualitative findings suggests that online support groups can bridge this gap as many group members reported feeling a sense of belonging, feeling less isolated and developing new friendships, which is in line with previous reviews[18,19]. However, it is important the group members also foster offline relationships as some may feel lonely when coming offline and others may focus on online connections as the expense of in-person connections. Furthermore, after reading others' experiences or advice, participants reported being motivated to keep up with self-management, change their behaviour and adopt new behaviours (e.g., changing their diet or purchasing assistive devices). This is in line with previous reviews reporting that social networking sites, peer support and online support groups may be effective for changing health behaviours[16,19,132]. For treatment decision-making (including treatment confidence) and empowerment, participants reported that sharing personal experiences and information helped them assess the benefits and side-effects of treatment, which in turn, helped them make decisions about their treatment and feel empowered. However, the benefits of the decisions may be context-dependent and vary according to the revised treatment option and group preferences[133]. For example, if a particular group encourages behaviours or treatments that could be damaging this could have a negative effect, as has been highlighted in the eating disorder literature comparing pro-eating disorder groups to pro-recovery groups[134]. It is also important to be cautious of misinformation and anecdotal evidence that may occur within online support groups. For example, whilst a particular treatment may be successful for one individual this is not to say it will work in someone else. As a result, many recommend speaking to a healthcare professional or conducting your own research before making changes[46].

However, some participants reported leaving the groups if their needs were not met, or they did not feel close with other group members. This highlights the importance of exploring the available online support groups to identify the ones that align with one's needs and values[46]. Quantitative, and experimental, findings were less likely to report changes in loneliness, friendships, or behaviour change, but this could be due to the research design whereby participants are aware of the short duration of the study or it could be that quantitative measures do not reflect the experienced behaviours and connections. As the cross-sectional studies for adjustment and social wellbeing are naturalistic and mostly qualitative, they provide an insight into the effects of online support groups used by participants in their day-to-day lives, but they cannot establish cause and effect, nor analyse outcomes over time.

For symptoms and functioning, depression, coping, quality of life, treatment adherence and self-efficacy the findings had either a positive or no effect, suggesting that whilst online support groups are unlikely to worsen these health outcomes, they may not always improve them. This supports a previous review which found mixed effects of various types of peer support in care settings on physical health outcomes[135] and partially supports a review that found computer-mediated support being associated with less depression and greater quality of life[20]. As people with chronic conditions are more likely to develop depression[136] and the physical symptoms are a key component affecting patients' day-to-day life[137], alternative support from a healthcare professional should be sought. It is also important to note that, compared to adjustment and social wellbeing, studies measuring these four outcomes mostly used quantitative and (quasi-)experimental measures which may also explain the findings. For example, the measures used may not reflect the experiences of those with a chronic condition or the experimental nature of the study may not lead to changes in health outcomes, either due to moderation, style or study duration.

There is also the potential for some health outcomes to worsen after engaging with an online support group. Whilst some studies reported either no change or improvements in anxiety and broad mental health and wellbeing, participants in a similar number of studies described increases in anxiety and feelings of frustration, sadness, and guilt. This may be, in part, due to the greater number of qualitative studies used to measure these outcomes, as they allow participants to provide greater insight into their experiences. However, it is also likely that online support groups can simultaneously help and hinder mental health and may be dependent on various factors, such as users' mood when engaging with the groups, group content and external pressures[46]. As a result, participants should be aware of this potentially harmful effect and should be attentive to how they feel when using online support groups and take a break if they notice a deterioration.

### What are the mechanisms by which online support groups affect the health and wellbeing of individuals with a chronic condition?

Health and wellbeing outcomes were influenced by giving and receiving support, and sharing experiences and social comparison, which supports the extant literature[16,138]. These findings also partially map onto the SCENA model with regards to connection, exploration and narration[25]. This review distinguishes between the differential role of informational and emotional support for health. Indeed, whilst informational support may aid physical health, adaptive coping and behaviour change and decision-making, emotional support may improve wellbeing, anxiety, illness acceptance and make individuals feel less alone, particularly in the absence of other care. The findings also suggest that giving emotional support may aid positive re-framing whereas receiving support may help with depression. However, whilst this supports a previous review of mechanisms of different types of peer support, which reported that helping others enabled peers to find meaning in their own chronic condition[129], multiple studies in this review reported that helping others was often not associated with outcome measures, including self-esteem, coping, optimism and social wellbeing. This may be due to the outcome measure or it could be attributed to the duration of participants' illness as they may be more likely to benefit from helping others at a later stage of their illness journey[46].

Qualitative studies suggesting that reading others' experiences can also positively influence physical health, adjustment, and behaviour and decision-making. However, quantitative measures did not find any effect on optimism and hope, potentially due to the way in which these measures were operationalised. Furthermore, if group members engage in positive comparison strategies whilst reading these experiences, it may help them feel less alone, normalise the condition, view their condition more positively, support meaning making, and put experiences put into 'proportion'[75,102]. This provides evidence for social comparison theory, which suggests that in order to evaluate oneself, people often compare to others[139]. This is typically done under uncertainty[139], which is often the case for people experiencing a chronic condition, particularly novel or under-researched conditions such

as Long Covid. However, reading others' experiences can also negatively influence broad mental health, anxiety and distress, particularly when posts are negatively oriented or include worse symptoms or experiences, as readers feel upset or guilty. Similarly, if group members engage in negative comparison strategies (e.g., feeling frustrated at others doing better or anxious of people being worse) then they may have a negative effect on mental health, so it is important that individuals draw inspiration from other group members rather than dwelling on negative aspects of comparison. Therefore, it is important for group members to be aware of these potential negative outcomes when choosing which posts to engage with.

Health outcomes may also differ depending on usage (e.g., level of engagement, intensity of use and membership duration) and group characteristics (e.g., moderated vs unmoderated and synchronous vs asynchronous). For most health outcomes, the included studies suggest that they are not impacted by the extent, or intensity, to which group members engage, but for feeling less alone, feeling more understood, and enhanced social wellbeing it may be beneficial to engage more actively and frequently. However as most of the studies were cross-sectional, it may be that individuals who already have these positive outcomes engage more with the groups. Also, a limited number of studies explored each outcome and characteristic, often with conflicting findings or different definitions and measurements, which makes it difficult to identify the optimal level of interaction with online support groups.

It is argued that different group features may afford different benefits and may depend on individual preferences[27]. Most of the studies in this review explored asynchronous groups, such as discussion forums, Facebook groups, or email lists, although one cross-sectional study found that video-based groups foster social wellbeing, compared to large text-based groups which aid informational support[46]. Synchronous groups were only explored in seven papers, most of which were (quasi)experimental, and the quantitative findings of these experiments mostly reported no effects, compared to control groups, on health outcomes. However, it is not possible to establish whether this was due to the design of the support group or the experimental, or quantitative, nature of the study, particularly as the quantitative findings of (quasi)experimental studies with asynchronous groups mostly reported similar results and the qualitative findings of both synchronous and asynchronous groups were more nuanced. Future research should explore this more rigorously.

Furthermore, studies comparing face-to-face and online groups found that both groups have similar influences on the health outcomes of group members. Two papers (with the same participants) compared unmoderated and moderated groups, and another two compared face-to-face and online groups, and found similar improvements in both groups. This is in line with a study which found no differences in depressive symptoms between participants allocated to a moderated or peer-led online support group[88]. However, it is not possible to generalise to other moderated groups, as groups can be moderated by researchers, peers or psychologists and can vary in activity from approving posts to actively guiding the conversation.

**Implications.** Living with a chronic condition can have various consequences on health and wellbeing, with many turning to online support groups to support these health outcomes. This review can be used by clinicians, online support group administrators and those with a chronic condition to optimise their experience of using online support groups. The following recommendations can be made based on this review: (i) As many health outcomes were not affected by level and intensity of engagement, group members can engage with the groups at their own pace without harming their health; (ii) Online support groups may be able to bridge the decline in offline relationships that can occur with the diagnosis of a chronic condition, but it is important to not do this at the expense of offline relationships; (iii) If group members are looking to make a behavioural change or find support with treatment decision-making, they may benefit from informational support, but should also conduct their own research or speak to a healthcare professional; (iv) If individuals do not know anyone else with their condition, seeking

emotional support from an online support group may help them feel less alone and more understood; (v) Learning of others' experiences, particularly those who are successfully managing the condition, can support illness acceptance and feeling 'normal', particularly for conditions with increased uncertainty; and vi) Individuals should be aware that online support groups have the potential to increase distress, anxiety and negative emotions, so it is important that they avoid negatively oriented posts and negative comparison strategies and take a break from groups if their mental health begins to decline. Whilst considering these recommendations, it is important to be aware of the limitations of this review and the included studies. It is also important to consider individual differences that may also affect experiences with online support groups.

**Limitations and future research.** Limitations of the studies included and of the review itself should be acknowledged. First, the quality of the studies was satisfactory. Most studies were cross-sectional, and survey findings were particularly descriptive, therefore it is not possible to identify a causal relationship between use of the online support groups and health and wellbeing. Also, some naturalistic studies did not include descriptions of the groups used, or whether participants were members of multiple groups thus making it difficult to extrapolate the findings. Indeed, it is possible that members used multiple online support groups, either for the same condition or to support them with multiple conditions, and that group values or content varied between groups, with each group potentially influencing health and wellbeing differently [115]. Most of the groups were also asynchronous, so it is not necessarily possible to extrapolate to synchronous groups, particularly video-based groups. The majority of included studies are also susceptible to selection bias, therefore it is possible that the samples do not reflect the wider population of either the online support groups or the chronic condition. When researching existing online support groups, researchers should endeavour to report as much detail as possible, such as whether members use multiple groups, their engagement level, and group features. Moreover, many of the studies also included mostly White and married participants, so these findings may not extrapolate to other demographics. This is important as chronic conditions may be more prevalent in deprived groups[140] and there may be different support needs between married and single participants[141]. When considering the review itself, it is possible that some studies were not identified within the search. There are also many offline factors, such as offline support and symptom severity, that may also underly any effects of online support groups on health and wellbeing[67], which were beyond the scope of this review. Finally, most studies were conducted in populations with cancer, which may skew the findings as there are considerable differences between the available formal support for cancer patients compared to conditions such as Myalgic Encephalomyelitis/Chronic fatigue syndrome or Long Covid[142].

There is scope for further research, particularly regarding the effects of different group features, such as group size, composition, platform, duration and moderators[143]. Future research should compare different levels of these features to identify the optimal set-up of these features (e.g., video- or text-based) and the most suitable type of moderator (e.g., peer or healthcare professional). Similarly, there were limited, and sometimes conflicting, findings for usage characteristics so it is important for studies to formally define active and passive users and further explore how this influences health outcomes. As most studies included in this review were cross-sectional, future research should also consider a longitudinal design to see if such effects were sustained over time and to identify possible spill-over effects changes.

**Conclusions.** This review synthesised findings on 25 health outcomes on the effects of online support groups for people with chronic conditions and suggests that online support groups broadly have a positive effect on social wellbeing (e.g., feeling connected to others and less isolated), behaviour (e.g., adopting positive behaviours), and adjustment (e.g.,

illness acceptance, identity, and feeling understood). For physical health, the findings suggest a positive influence on pain but a mixed result for symptoms and functioning. In terms of mental health, online support groups may have a positive or negative impact on outcomes, such as anxiety and emotional or psychological wellbeing, and this will depend on group content and comparison strategies.

## Data availability
Data and materials used for this review are available in Tables 1–12 in the Supplementary File.

## Code availability
No custom code was used in data collection or analysis.

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

## Acknowledgements

The authors would like to thank Caroline De Brun, from UKHSA, and Tom Kowalczyk, from University of Sussex, for their support and guidance with database searches and providing feedback on initial versions of the search strategy. This study was funded by the National Institute for Health and Care Research Health Protection Research Unit (NIHR HPRU) in Emergency Preparedness and Response, a partnership between the UK Health Security Agency, King's College London and the University of East Anglia. The funders had no role in study design, data collection and analysis, decision to publish or preparation of the manuscript. The views expressed are those of the author(s) and not necessarily those of the NIHR, UKHSA or the Department of Health and Social Care.

## Author contributions

Freya Mills: Conceptualization, Methodology, Formal analysis, Investigation, Writing – Original Draft, Project administration. John Drury: Conceptualization, Methodology, Formal analysis, Writing - Review & Editing, Supervision Charlotte E Hall: Validation, Writing - Review & Editing Dale Weston: Methodology, Writing - Review & Editing, Supervision Charles Symons: Methodology, Writing - Review & Editing, Supervision Richard Amlôt: Writing - Review & Editing, Supervision Holly Carter: Conceptualization, Methodology, Formal analysis, Writing - Review & Editing, Supervision.

## Competing interests

The authors declare no competing interests.
