## [Transparent Peer Review file · Communications Psychology]

A mixed studies systematic review on the health and wellbeing effects, and underlying mechanisms, of online support groups for chronic conditions

Corresponding Author: Ms Freya Mills

Version 0:

Decision Letter: first round

Dear Ms Mills,

Thank you for your patience during the peer-review process. Your manuscript titled "A mixed studies systematic review on the health and wellbeing effects, and underlying mechanisms, of online support groups for chronic conditions" has now been seen by 2 reviewers, whose comments are appended below. You will see that they find your work of some potential interest. The Reviewers are enthusiastic about the aims of the work and their constructive reports are supportive of your manuscript. Nonetheless, there are quite substantial concerns that must be addressed within the reviewer reports and from an editorial perspective. We therefore cannot accept the manuscript for publication, but would be interested in considering a revised version that fully addresses these serious concerns.

Should additional work allow you to address these criticisms, we would be happy to look at a substantially revised manuscript. If you choose to take up this option, please highlight all changes in the manuscript text file, and provide a detailed point-by-point reply to the reviewers.

Editorially, we ask that the literature search is expanded to include the most recent publications (all papers up until August 2024). As this undertaking will require an additional round of data analysis, we ask that you implement an improved screening process in response to the criticism voiced by Reviewer for all eventually included publications (esp points 11 and 14). Relatedly, we also ask that you improve the transparency of reporting and analysis to alleviate the key concerns voiced by Reviewer #2 which may also require additional analysis (at least descriptive analyses).

We note that one of the 3 preregistered research questions is not presented in the present manuscript. Please note that as per our policy for preregistration, "authors must disclose all deviations from the preregistered protocol and explain the rationale for deviation (e.g., flaw, feasibility, suboptimality). In cases of deviation from the preregistered analysis plan for reasons other than fundamental flaw or feasibility, the originally planned analyses must also be reported."

Finally, we highlight that systematic reviews in the journal should be accompanied by a meta-analysis, unless the difference between study measures or number of studies renders this type of quantitative analysis unfeasible. Please respond to this issue in revision and consider inclusion of mini-meta analyses where appropriate and feasible.

I am attaching a checklist that details critical reporting requirements for the revised manuscript. Please attend to each item and ensure your manuscript is fully compliant. We are requesting that your manuscript aligns with these requirements as this facilitates the evaluation of your manuscript, reducing delays in re-review and potential future acceptance. If your revised manuscript is not aligned with these requests on major issues, such as those concerning statistics, it may be returned to you for further revisions without re-review. Additional information can be found in our style and formatting guide Communications Psychology formatting guide.

If the revision process takes significantly longer than five months, we will be happy to reconsider your paper at a later date, provided it still presents a significant contribution to the literature at that stage.

Please use the following link to submit your

- revised manuscript,
- point-by-point response to the referees' comments,
- cover letter (as a separate document),
- the Editorial Policy Checklist (see below),
- the Reporting Summary (see below), and
- the completed Editorial Request Table (attached):

Link Redacted

Thank you for the opportunity to review your work.

Best regards,

Marika

Marika Schiffer, PhD
Chief Editor
Communications Psychology

REVIEWER EXPERTISE:

Reviewer #1 systematic reviews, online health interventions
Reviewer #2 systematic reviews, online health interventions

REVIEWER REPORTS:

Reviewer #1 (Remarks to the Author):

Dear Editor,

Thank you for the opportunity to review the manuscript titled "A mixed studies systematic review on the health and wellbeing effects, and underlying mechanisms, of online support groups for chronic conditions".

I was particularly pleased to receive this manuscript as such a systematic review is long overdue and is urgently needed in the field. It is with optimism and positive intent that I reviewed this manuscript.

Overall, I strongly believe there is a place for this study in the literature and I would recommend it for publication. That said, I think there are several issues to consider and address before it is suitable for publication.

My comments are as follows:

1. In the abstract the authors make reference to 'hearing' and this implies that any negative impacts are only in those online support groups which offer an audio function. I don't believe this is the case and would ask the authors to revisit the choice of word here and perhaps replace it with '...particularly when exposed to other group members' difficult experiences".
2. The last line in the abstract 'Research comparing different types of support groups is needed". This is quite a vague statement, and the authors may wish to be more specific as to what they mean by 'types'.
3. Para 1: It is more accurate to say that approximately half of the population in the UK are living with at least one chronic condition. As it is currently worded, the reader might think you are saying approximately half the population live only with one chronic condition.
4. Para 1: The term 'kill' is uncomfortable. Could this be changed to 'die from'?
5. Para 1: Is the reference to 2 million people experiencing symptoms of long covid – UK specific figure? Global figure? This needs to be clarified.
6. Para 2: I think a brief definition of what an online support group is would be helpful in this paragraph. It needn't be lengthy

but enough to explain to the reader what they are. From this, they can continue to explain that they are underpinned by various platforms.

7. Para 2: It would be helpful if the authors could include a statement which points to the increasing numbers of online support groups which exist and the increasing numbers of people who are accessing them for information, advice and support. This number is likely to have increased because of the covid-19 pandemic – and is an important contextual point.
8. The Introduction does acknowledge that online support groups can be underpinned by various platforms. However, I feel the authors have underplayed the importance of platforms and their different affordances. For example, the SCENA model developed by Merolli provides a strong indication that the impact of online support groups could be through the interaction between the person and the platform and that not all platforms have the same affordances. I would encourage the authors to consider this issue and whether they could make a stronger reference to the importance of platform (and then again link this in the findings). For example – the experience and impacts of an asynchronous discussion forum may well be different to that of a synchronous audio/video enabled support group.
9. The 'Search criteria' lists Google Scholar as a database. It is not considered a database but rather an academic search engine.
10. It is not clear why the authors chose to limit their search to studies only in English? A brief explanation would be useful.
11. Given the size of the review team, it is surprising that only 10% of titles and abstracts were screened and 5% of full texts. In addition, the quality appraisal appears to have been undertaken by only one person – was anyone else involved to enhance reliability of the assessment? This raises concerns about the rigour of the review and the potential for bias and error.
12. In the 'Study characteristics' section, is it possible to include a statement reporting the range in years from the first study to the most recent and/or something to give the reader an idea as to when this area of research began and how there has been a significant increase in studies considering online support groups.
13. There is a type in Table 2. It is Crohn's not Chron's. Also, please amend 'Polycystic ovaries' to 'Polycystic Ovary Syndrome' - to be consistent with the description of other syndromes in this table.
14. How was the location of the study determined? I am surprised that only 7 were in the UK. So much so that I went through the list of studies references in Table 2 and counted more than 7 which were conducted in the UK. I think it could be useful to be clearer about the location of the study as determined by the country of the author (or corresponding author). However, if the authors are interested in the location of the group members, then this is a different issue and one which is altogether more complicated to address as many of the studies included in this review drew upon an international group of members.
15. Table 5 lacks detail and could benefit from some footnotes providing a succinct definition: i) closed group; ii) local group.
16. On page 29 (and other pages later), reference is made to an online 'community' or 'communities' – but the language thus far has been 'group'. It might be worth either changing community to group OR saying somewhere in the Introduction that OSGs are also known as online communities etc – so the reader appreciates there may be differences in language used across studies. For those readers who are not familiar with this area, the terms 'online group' and 'online community' might mean different things.
17. In the 'Limitations and future research' section, the authors correctly acknowledge that an individual may belong to more than one group. However, what is absent in this section is an acknowledgement that many people live with more than one chronic condition, and this may be the reason for multiple group members.
18. In the Discussion section, I think much more needs to be said about the quality assessment and what this all means for this area of research.
19. Linking to an earlier point, I think the Discussion needs to more explicitly consider the role of platform and interaction with the person when considering outcomes. This can easily be addressed by inclusion of 2-3 lines of text.
20. The manuscript requires a thorough proof-read, especially the Discussion section.

And finally...

I would like to positively commend the team on their well-developed and considered search strategy. It was very comprehensive.

Reviewer #2 (Remarks to the Author):

Thank you for the opportunity to review and comment on your research. I find this topic to be very interesting and timely. I hope you find my feedback useful in your revisions.

Abstract

The abstract is compressive and clearly written.

Introduction

It might be useful to report how many people with chronic conditions use online support groups.

There also needs to be some defining of different types of online groups, RCTs and peer-to-peer, for example. Add context to why the make up of these groups are different and the outcomes of each.

Line 82 – missing word after “provide strong...”

Line 92 – choose one – suggest or highlight

Line 95 – give e.g. of group features

Line 97 – Expand upon, “Meanwhile, larger groups are reported to be positively associated with quality of life scores but negatively associated with social support.”

Line 100 – “...support group content (e.g., support)...” > what is the difference?

Method

Line 129 – Define “grey literature” briefly.

Would be helpful to have a breakdown of the different methodologies used by the studies included in the review, and the different sample sizes.

Results

The Tables are helpful in comparing the articles included in the review.

I don't have any suggestions on how to improve the presentation of results, but it's difficult to sort through as it is written. It could be, this is just the best way to present the results, and to draw conclusions from the articles.

Discussion

Line 768 – The first sentence is too broad. Can the authors be more specific in this first sentence?

Line 783 – Refers to peer support, and again – this needs to be differentiated from formal intervention studies.

For RQ1, it is important to not overstate the effects of the studies in the sample.

Overall, I think this paper is good and needed, BUT. I would like to see the types of communities parsed out further because there's some conflating peer-to-peer with intervention groups here.

EDITORIAL POLICIES

We ask that you ensure your manuscript complies with our editorial policies and reporting requirements.

To that end, we require revised manuscripts to be accompanied by two completed items: a reporting summary that collects information on study design and procedure, and an editorial policy checklist that verifies compliance with all required editorial policies

- <https://www.nature.com/documents/nr-reporting-summary.zip>>Nature Research Reporting Summary
- <https://www.nature.com/documents/nr-editorial-policy-checklist.pdf>>Editorial Policy Checklist

All points on the policy checklist must be addressed. Your revised manuscript can only be sent back to the referees if these checklists are completed and uploaded with the revision.

Notes: If you have submitted a Stage 1 Registered Report, Review, Primer, Comment, or Perspective you do not need to submit these forms. If you have already submitted these forms, you may disregard this request.

** Visit Nature Research's author and referees' website at <http://www.nature.com/authors>>www.nature.com/authors for information about policies, services and author benefits**

Version 1:

Decision Letter: second round

Dear Ms Mills,

Your manuscript titled "A mixed studies systematic review on the health and wellbeing effects, and underlying mechanisms, of online support groups for chronic conditions" has now been seen by our reviewers, whose comments appear below. In light of their advice I am delighted to say that we are happy, in principle, to publish a suitably revised version in Communications Psychology.

We therefore invite you to revise your paper one last time to address the remaining concerns of our reviewers and a list of editorial requests. At the same time we ask that you edit your manuscript to comply with our format requirements and to maximise the accessibility and therefore the impact of your work.

EDITORIAL REQUESTS:

SUBMISSION INFORMATION:

OPEN ACCESS:

*** TRANSPARENT PEER REVIEW:** Communications Psychology uses a transparent peer review system. On author request, confidential information and data can be removed from the published reviewer reports and rebuttal letters prior to publication. If you are concerned about the release of confidential data, please let us know specifically what information you would like to have removed. Please note that we cannot incorporate redactions for any other reasons.

*** CODE AVAILABILITY:** All Communications Psychology manuscripts must include a section titled "Code Availability" at the end of the methods section. We require that the custom analysis code supporting your conclusions is made available in a publicly accessible repository at this stage; please choose a repository that generates a digital object identifier (DOI) for the code; the link to the repository and the DOI must be included in the Code Availability statement. Publication as Supplementary Information will not suffice.

*** DATA AVAILABILITY:**

All Communications Psychology manuscripts must include a section titled "Data Availability" at the end of the Methods

section. More information on this policy, is available in the Editorial Requests Table and at <http://www.nature.com/authors/policies/data/data-availability-statements-data-citations.pdf>.

Link Redacted

Best regards,

Marike Schiffer

Marike Schiffer, PhD
Chief Editor
Communications Psychology

REVIEWERS' EXPERTISE:

Reviewer #1

Reviewer #2

REVIEWERS' COMMENTS:

Reviewer #1 (Remarks to the Author):

I would like to thank the authors for their comprehensive and well considered responses to the various points raised during peer review round 1. I very much appreciate the additional thinking, clarification and development of the manuscript. I consider all the points I raised addressed in a satisfactory way and have no additional points to raise at this time.

I would recommend this excellent manuscript for publication and thank the team for their herculean efforts to undertake this systematic review and I look forward to seeing it published. It will be a great asset to the literature. Thank you.

Response to Reviewer Comments

RE Manuscript ID: COMMSPSYCHOL-24-0400

Title: A mixed studies systematic review on the health and wellbeing effects, and underlying mechanisms, of online support groups for chronic conditions

Dear Dr Schiffe,

Thank you for your consideration of our paper. We would like to thank your reviewers for their constructive comments on our manuscript. We have made the revisions requested and hope that it is now acceptable for publication.

Point by point responses to the reviewer comments are given below. The reviewer comment is in bold with our response below. Changes made to the manuscript are presented with indented extracts. The following changes have also been made in the manuscript and are in tracked changes.

Reviewer #1 (Remarks to the Author):

Dear Editor,

Thank you for the opportunity to review the manuscript titled “A mixed studies systematic review on the health and wellbeing effects, and underlying mechanisms, of online support groups for chronic conditions”.

I was particularly pleased to receive this manuscript as such a systematic review is long overdue and is urgently needed in the field. It is with optimism and positive intent that I reviewed this manuscript.

Overall, I strongly believe there is a place for this study in the literature and I would recommend it for publication. That said, I think there are several issues to consider and address before it is suitable for publication.

Thank you for the positive comments regarding the review. We hope it will provide the needed overview of the landscape of the existing research into online support groups.

My comments are as follows:

1. In the abstract the authors make reference to ‘hearing’ and this implies that any negative impacts are only in those online support groups which offer an audio function. I don’t believe this is the case and would ask the authors to revisit the choice of word here and perhaps replace it with ‘...particularly when exposed to other group members’ difficult experiences”.

We agree that the term ‘hearing’ could be misconstrued. We have changed this term to “exposed”, as suggested.

2. The last line in the abstract ‘Research comparing different types of support groups is needed”. This is quite a vague statement, and the authors may wish to be more specific as to what they mean by ‘types’.

We have added some more detail to the abstract and clarify what we mean by ‘types’. We now say:

Research comparing different group features, such as platforms, size, and duration is needed.

3. Para 1: It is more accurate to say that approximately half of the population in the UK are living with at least one chronic condition. As it is currently worded, the reader might think you are saying approximately half the population live only with one chronic condition.

We understand how this statistic may be misinterpreted. We have changed this to the following:

Almost half the UK population reported living with at least one long-standing health condition in 2020

4. Para 1: The term ‘kill’ is uncomfortable. Could this be changed to ‘die from’?

We agree that this a strong and uncomfortable word. We have changed this to:

Globally, 41 million per year are estimated to die from a chronic condition

5. Para 1: Is the reference to 2 million people experiencing symptoms of long covid – UK specific figure? Global figure? This needs to be clarified.

We have now clarified this statistic to specify that it is in England and Scotland.

6. Para 2: I think a brief definition of what an online support group is would be helpful in this paragraph. It needn't be lengthy but enough to explain to the reader what they are. From this, they can continue to explain that they are underpinned by various platforms.

We agree that a definition will be helpful for readers. We have added the following definitions:

Online support groups are “online services with features that enable members to communicate with each other” [11]; they have an underlying premise that peers offer meaningful support due to the shared experience of a particular life event [12]

7. Para 2: It would be helpful if the authors could include a statement which points to the increasing numbers of online support groups which exist and the increasing numbers of people who are accessing them for information, advice and support. This number is likely to have increased because of the covid-19 pandemic – and is an important contextual point.

We agree that it is useful to provide this context and have now added the following:

The growing need for online support groups is showcased by the large membership of many groups. For example, at the time of writing a diabetes Facebook group has reached 102,000 members in four years, 202 new members in the last week, and has 202 posts per month [13], and a Long Covid support group has reached 66,000 members, 96 members in the last week and has 2,000 posts in the last month [14].

8. The Introduction does acknowledge that online support groups can be underpinned by various platforms. However, I feel the authors have underplayed the importance of platforms and their different affordances. For example, the SCENA model developed by Merolli provides a strong indication that the impact of online support groups could be through the interaction between the person and the platform and that not all platforms have the same affordances. I would encourage the authors to consider this

issue and whether they could make a stronger reference to the importance of platform (and then again link this in the findings). For example – the experience and impacts of an asynchronous discussion forum may well be different to that of a synchronous audio/video enabled support group.

We agree that the platforms used by online support groups can affect the experiences. We have now addressed this throughout the paper. For example, in the introduction we now say:

Furthermore, the SCENA Model of Therapeutic Affordances of Social Media [25] has also been applied to online support groups [26], and suggests that such groups may afford self-presentation (*managing how one presents themselves online*), connection (*connecting with, and supporting, others*), exploration (*seeking information and improving knowledge*), narration (*exchanging experiences*) and adaptation (*adapting self-management needs in relation to health status*). Due to the variety of platforms used for online support groups (e.g., video- or text-based), as well as the different ways of engaging with the groups (i.e., being a passive or active member of the group). It has been argued that the different types of platforms enable the benefits afforded by online support groups in different ways [27]. For example, exploration may be easier in text-based groups where this an archive of information.

In the results section, we have included the types of platforms (where possible) used within the studies. For example, we now say:

75 papers looked at asynchronous groups (e.g., discussion forums, Facebook groups, WhatsApp groups, email lists), seven papers (of which six were experimental) looked at synchronous groups (e.g., real time text-based chat groups, or video or teleconference calls), and four explored a combination of both synchronous and asynchronous groups.

Also, in the discussion we mention the SCENA model in relation to research question 2, and we further discuss the role of different platforms:

Health and wellbeing outcomes were influenced by giving and receiving support, sharing experiences and social comparison, which supports the extant literature [12, 124]. The outcomes also partially map onto the SCENA model with regards to connection, exploration and narration [25].

It is argued that different group features may afford different benefits and may depend on individual preferences [27]. Most of the studies in this review explored asynchronous groups, such as discussion forums, Facebook groups, or email lists, although one cross-sectional study found that video-based groups foster social wellbeing, compared to large text-based groups which aid informational support [136]. Synchronous groups were only explored in seven papers, most of which were (quasi)experimental and the quantitative findings of these experiments mostly reported no effects, compared to control groups, on health outcomes. However, it is not possible to establish whether this was due to the design of the support group or the experimental, or quantitative, nature of the study, particularly as the quantitative findings of (quasi)experimental studies with asynchronous groups mostly reported similar findings and the qualitative findings of both synchronous and asynchronous groups were more nuanced. Future research should explore this more rigorously.

In the limitation section we have also explained that most of the groups were asynchronous so it's not necessarily possible to extrapolate, particularly to video-based groups.

9. The 'Search criteria' lists Google Scholar as a database. It is not considered a database but rather an academic search engine.

We have changed the text to call Google Scholar an academic search engine.

10. It is not clear why the authors chose to limit their search to studies only in English? A brief explanation would be useful.

This limitation is due to the language abilities of the research team. We have explained this in the paper by saying:

Studies from any country were included, if they were published in English, due to the languages spoken by the research team.

11. Given the size of the review team, it is surprising that only 10% of titles and abstracts were screened and 5% of full texts. In addition, the quality appraisal appears to have been undertaken by only one person – was anyone else involved to enhance reliability of the assessment? This raises concerns about the rigour of the review and the potential for bias and error.

We appreciate the concerns relating to the rigour of the review and the potential for bias and error. We have now updated our screening practices. We consulted our library services for guidance with this. Where feasible, we followed the guidance outlined by Garritty et al (2020). In our updated screening, the third author screened 20% of titles and abstracts and all of the excluded full texts. Although one author conducted the quality appraisal, the quality of the studies was discussed with the research team, where questions were prompted (e.g., in relation to the heterogeneity of scales used for each outcome) and the impact of this quality was discussed (e.g., in relation to the cross-sectional design).

12. In the 'Study characteristics' section, is it possible to include a statement reporting the range in years from the first study to the most recent and/or something to give the reader an idea as to when this area of research began and how there has been a significant increase in studies considering online support groups.

We have now added the years that the earliest and most recent studies were published, and we have included the years with the largest number of published studies:

Studies were published between 2002 and 2024. The years with the largest number of published studies were 2021 (n = 10), 2022 (n = 11), and 2024 (n = 9).

13. There is a type in Table 2. It is Crohn's not Chron's. Also, please amend 'Polycystic ovaries' to 'Polycystic Ovary Syndrome' – to be consistent with the description of other syndromes in this table.

Thank you for identifying these errors. We have made these changes.

14. How was the location of the study determined? I am surprised that only 7 were in the UK. So much so that I went through the list of studies references in Table 2 and counted more than 7 which were conducted in the UK. I think it could be useful to be clearer about the location of the study as determined by the country of the author (or corresponding author). However, if the authors are interested in the location of the group members, then this is a different issue and one which is altogether more complicated to address as many of the studies included in this review drew upon an international group of members.

We have updated these details to include both first author location and participant location (where possible). Specifically, we now say:

Most studies were conducted by authors based in the USA (n = 47), followed by the UK (n = 18), Netherlands (n = 7) and Canada (n = 4). 30 studies did not report participant location. Amongst those that did, most participants were based in the USA (n = 18), the Netherlands (n = 7), the UK (n = 9) and China (n = 5) or were international, but with a high proportion of participants in the USA (n = 7). These numbers were identified based on inclusion criteria or recruitment details (e.g., recruited via a specific hospital, university or a location-specific support group).

15. Table 5 lacks detail and could benefit from some footnotes providing a succinct definition: i) closed group; ii) local group.

We have added in some explanations for these groups. For example, we now say “closed groups (groups that require permission to join)” and “local groups (groups with a focus on a local area, rather than groups with an international or non-specific geographical focus)”

16. On page 29 (and other pages later), reference is made to an online ‘community’ or ‘communities’ – but the language thus far has been ‘group’. It might be worth either changing community to group OR saying somewhere in the Introduction that OSGs are also known as online communities etc – so the reader appreciates there may be differences in language used across studies. For those readers who are not familiar with this area, the terms ‘online group’ and ‘online community’ might mean different things.

We understand that there may be potential confusion here. In the introduction, we have added alternative names for online support groups: “Online support groups, also referred to as ‘online communities’, ‘online support forums’, and ‘virtual support groups’”.

Within the results section we have aimed to use the language reported in the original papers, which is why there may be reference to ‘online communities’.

17. In the ‘Limitations and future research’ section, the authors correctly acknowledge that an individual may belong to more than one group. However, what is absent in this section is an acknowledgement that many people live with more than one chronic condition, and this may be the reason for multiple group members.

We agree that it is important to acknowledge that people may be living with more than one chronic condition. We have added the following to the discussion:

Also, it is possible that members used multiple online support groups, either for the same condition or to support them with multiple conditions, and that group values or content varied, with each group potentially influencing health and wellbeing differently

18. In the Discussion section, I think much more needs to be said about the quality assessment and what this all means for this area of research.

We have expanded the limitations section to discuss the quality of the research reviewed.

We now say:

Limitations of the studies included and of the review itself should be acknowledged. First, the quality of the studies was satisfactory, but as most studies were cross-sectional and the survey findings were particularly descriptive, it is not possible to identify a causal relationship between use of the online support groups and health and wellbeing. Also, some naturalistic studies did not include descriptions of the groups used, or whether participants were members of multiple groups thus making it difficult to extrapolate the findings. Furthermore, it is possible that members used multiple online support groups, either for the same condition or to support them with multiple conditions, and that group values or content varied, with each group potentially influencing health and wellbeing differently [115]. Most of the groups were also asynchronous, so it is not necessarily possible to extrapolate to synchronous groups, particularly video-based groups. The majority of included studies are also susceptible to selection bias, therefore it is possible that the samples do not reflect the wider population of either the online support groups or the chronic condition. These limitations are inherent with the study of online support groups and individuals with chronic conditions. When researching existing online support groups, researchers should endeavour to report as much detail as possible, such as whether members use multiple groups, their engagement level, and group features.

19. Linking to an earlier point, I think the Discussion needs to more explicitly consider the role of platform and interaction with the person when considering outcomes. This can easily be addressed by inclusion of 2-3 lines of text.

We have added some more details on this into the discussion, particularly in relation to synchronous and asynchronous groups.

It is argued that different group features may afford different benefits and may depend on individual preferences [27]. Most of the studies in the review were asynchronous groups, such as discussion forums, Facebook groups, or email lists, although one cross-sectional study found that video-based groups foster social wellbeing compared to large text-based groups which aid informational support [136]. Synchronous groups were only explored in seven papers, most of which were (quasi)experimental, and the quantitative findings of these experiments mostly reported no effects, compared to control groups, on health outcomes. However, it is not possible to establish whether this was due to the design of the support group or the experimental, or quantitative, nature of the study, particularly as the quantitative findings of (quasi)experimental studies with asynchronous groups mostly reported similar findings and the qualitative findings of both synchronous and asynchronous groups were more nuanced. Future research should explore this more rigorously.

20. The manuscript requires a thorough proof-read, especially the Discussion section.

We have been through the discussion and amended any typos or grammatical errors.

And finally...

I would like to positively commend the team on their well-developed and considered search strategy. It was very comprehensive.

Thank you for the kind feedback.

Reviewer #2 (Remarks to the Author):

Thank you for the opportunity to review and comment on your research. I find this topic to be very interesting and timely. I hope you find my feedback useful in your revisions.

Thank you for your comments, we are glad to hear that you find the topic interesting.

Abstract

The abstract is compressive and clearly written.

Thank you.

Introduction

It might be useful to report how many people with chronic conditions use online support groups.

We agree that this is useful information. We couldn't find an overall figure of how many people are using online support groups, but we have provided some context by giving the numbers of members in a Long Covid group and a diabetes support group. We've added the following text:

The need for online support groups is showcased by the large membership of many groups. For example, in the last four years, a diabetes Facebook group has reached 102,000 members, at the time of writing, 202 new members in the last week, and has 202 posts per month [13], and a Long Covid support group has reached 66,000 members, 96 members in the last week and has 2,000 posts in the last month [14].

There also needs to be some defining of different types of online groups, RCTs and peer-to-peer, for example. Add context to why they make up of these groups are different and the outcomes of each.

All online support groups within the review offer peer-to-peer support. We have now specified this in the introduction by defining online support groups as:

“online services with features that enable members to communicate with each other”; [11] they have an underlying premise that peers offer meaningful support [12].

Whilst some studies included in the review are RCTs, or quasi-experiments, peer-to-peer support occurs and is encouraged in all of these. Some studies may be professionally moderated (e.g., by a research team, a charity, or psychologist) and may have weekly topics to guide the group, but they will still have peer-to-peer support. Throughout the introduction we have referred to the variations in the make-up of online support groups, for example:

They may be created, and moderated, by peers (i.e., those with a direct lived experience of the condition), caregivers and health professionals.

Such groups can be synchronous via audio or video calls, or they can be asynchronous via social media platforms, such as Facebook groups and discussion boards, or via direct messages, such as in WhatsApp groups.

The results section provides context to the proportion of studies that are experimental and naturalistic, their moderation style, and the types of platforms used. We say:

The effects of online support groups were tested with a variety of methods with the most frequent being cross-sectional surveys (quantitative and qualitative; $n = 52$), cross-sectional interviews ($n = 24$) and quasi-experimental studies ($n = 16$). Experimental studies introduced participants to a new online support group, often created for the experiment whereas cross-sectional studies and longitudinal surveys were naturalistic as they typically assessed the impact of groups in which participants were already a member. Interventions lasted between 1 and 6 months, whilst the duration of support group membership in cross-sectional studies, when reported, ranged between less than 1 week to 15 years with reported mean duration being between 12 and 31 months. Groups created for the purpose of the research were mostly moderated by the researchers, psychologists, healthcare professionals or patient organisations, whereas studies exploring naturalistic groups often did not report how the group was moderated. 75 papers looked at asynchronous groups (e.g., discussion forums, Facebook groups, WhatsApp groups, email lists), seven papers (of which six were experimental) looked at synchronous groups (e.g., real time text-based chat groups, or video or teleconference calls), and four explored a combination of both synchronous and asynchronous groups.

The discussion also explores differences between experimental vs non-experimental groups, particularly in relation to synchronous and asynchronous groups. All of these groups included peer support:

It is argued that different group features may afford different benefits and may depend on individual preferences [27]. Most of the studies in the review were asynchronous groups, such as discussion forums, Facebook groups, or email lists, although one cross-sectional study found that video-based groups foster social wellbeing compared to large text-based groups which aid informational support [136]. Synchronous groups were only explored in seven papers, most of which were (quasi)experimental, and the quantitative findings of these experiments mostly

reported no effects, compared to control groups, on health outcomes. However, it is not possible to establish whether this was due to the design of the support group or the experimental, or quantitative, nature of the study, particularly as the quantitative findings of (quasi)experimental studies with asynchronous groups mostly reported similar results and the qualitative findings of both synchronous and asynchronous groups were more nuanced. Future research should explore this more rigorously.

Line 82 – missing word after “provide strong...”

Line 92 – choose one – suggest or highlight

Line 95 – give e.g. of group features

Thank you for noticing these errors; we have now corrected these.

Line 97 – Expand upon, “Meanwhile, larger groups are reported to be positively associated with quality of life scores but negatively associated with social support.”

As we have added more to this paragraph of the introduction in response to Reviewer 1, we have removed this sentence.

Line 100 – “...support group content (e.g., support)...” > what is the difference?

We have removed ‘(e.g., support)’ from the sentence.

Method

Line 129 – Define “grey literature” briefly.

We have added the following explanation for grey literature:

Grey literature searches were also conducted to identify any eligible reports not published via academic publishers, to ensure comprehensiveness,

Would be helpful to have a breakdown of the different methodologies used by the studies included in the review, and the different sample sizes.

We agree that it is useful to know the different methodologies used in this review. The results section details the most frequently used methodologies, as we say:

The effects of online support groups were tested with a variety of methods with the most frequent being cross-sectional surveys (quantitative and qualitative; n = 52), cross-sectional interviews (n = 24) and quasi-experimental studies (n = 16).

Table 3 also shows the distribution of different methodologies across each health outcome and throughout the results section the findings for each health outcome are grouped by research method.

Sample sizes for each study are included in the supplementary files, and we have added the following sentence to show the range of sample sizes:

Sample sizes of the included studies ranged from 1641 to 6 participants

Results

The Tables are helpful in comparing the articles included in the review.

Thank you, we are glad the tables were helpful.

I don't have any suggestions on how to improve the presentation of results, but it's difficult to sort through as it is written. It could be, this is just the best way to present the results, and to draw conclusions from the articles.

We appreciate that there is a lot of content. We have tried to break the findings down in the best way possible whilst ensuring that we reflect the differences across research methods for each outcome. We hope that by including the summary table and by organising the discussion by research method that it is clear the different effects of online support groups on health outcomes.

Discussion

Line 768 – The first sentence is too broad. Can the authors be more specific in this first sentence?

We have tried to make the first sentence more specific by saying:

This systematic review sought to investigate which health outcomes are influenced by participating in online support groups. We also sought to identify the factors influencing such outcomes.

Line 783 – Refers to peer support, and again – this needs to be differentiated from formal intervention studies.

We agree that there is a difference between cross-sectional studies, that are more likely to investigate naturalistic groups, and intervention studies. All of the studies included in the review are peer support, although some (particularly intervention studies) are also formally moderated to prompt this support.

When “peer support” is mentioned in the discussion, this is typically referring to other systematic reviews, or studies, that look at peer support more broadly and includes support across a variety of dimensions including one-to-one, groups, online, or in person. These studies have been included in the discussion to draw comparisons between the findings. Where social / informational / emotional support is mentioned, this is referring to the self-report data from the included studies and how they influence health outcomes.

We have tried to differentiate between types of online support groups in the discussion, but as few studies actively compare different support group features this is difficult. We have included the following sentences to draw upon synchronous vs asynchronous groups, experimental vs non-experimental, and moderated vs unmoderated groups.

It is argued that different group features may afford different benefits and may depend on individual preferences [27]. Most of the studies in the review were asynchronous groups, such as discussion forums, Facebook groups, or email lists, although one cross-sectional study found that video-based groups foster social wellbeing, compared to large text-based groups which aid informational support [136]. Synchronous groups were only explored in seven papers, most of which were (quasi)experimental and the quantitative findings of these experiments mostly reported no effects, compared to control groups, on health outcomes. However, it is not possible to establish whether this was due to the design of the support group or the experimental, or quantitative, nature of the study, particularly as the quantitative findings of (quasi)experimental studies with asynchronous groups mostly reported similar findings and the qualitative findings of both synchronous and asynchronous groups were more nuanced. Future research should explore this more rigorously.

For RQ1, it is important to not overstate the effects of the studies in the sample.

We agree that it is important to not overstate the effects. We have amended the language to reflect the participants' experiences, rather than making conclusive statements. For example:

Participants reported that online support groups helped them find meaning, feel “normal” and either re-discover their old sense of self or discover a new identity, which is particularly important as those with chronic conditions often experience a loss of personal identity

Overall, I think this paper is good and needed, BUT. I would like to see the types of communities parsed out further because there's some conflating peer-to-peer with intervention groups here.

Thank you for agreeing that this is a needed paper in this area. We hope we have addressed your concerns regarding the different communities. All of the studies included in the review are peer support, although there are different levels of moderation which may prompt this support and we agree that the experiences may differ between naturalistic and intervention groups. Throughout the results section we have tried to be clear on the research method that each study comes from by including statements such as “An RCT found...”. In experimental studies, we have also tried to be clear about the details of the online support groups, for example: “An RCT comparing a peer-led Facebook group plus online education to education alone found no differences in empowerment at 3 or 6 months”. We have also tried to explore this in the discussion with the experts included above.